# WILDCHAT-50M: A Deep Dive Into the Role of Synthetic Data in Post-Training

**Benjamin Feuer** [1]   **Chinmay Hegde** [1]

## Abstract

Language model (LLM) post-training, from DPO to distillation, can refine behaviors and unlock new skills, but the open science supporting these post-training techniques is still in its infancy. One limiting factor has been the difficulty of conducting large-scale comparative analyses of synthetic data generating models and LLM judges. To close this gap, we introduce WILDCHAT-50M, the largest public chat dataset to date. We extend the existing WildChat dataset to include responses not only from GPT, but from over 50 different open-weight models, ranging in size from 0.5B to 104B parameters. We conduct an extensive comparative analysis and demonstrate the potential of this dataset by creating RE-WILD, our own public SFT mix, which outperforms the recent Tulu-3 SFT mixture from Allen AI with only 40% as many samples. Our dataset, samples and code are available at https://github.com/penfever/wildchat-50m.

## 1. Introduction

Large language model (LLM) post-training encompasses a broad suite of algorithmic techniques, and is an active area of current research. Improvements in LLM post-training have led to many breakthrough accomplishments, ranging from recent developments in test-time scaling from OpenAI and Deepseek (OpenAI et al., 2024; DeepSeek-AI et al., 2025) to new algorithms for efficiently aligning LLMs to human preferences (Rafailov et al., 2024). All of them rely on synthetic data during post-training, sometimes in the form of judgments through LLM judges or pairwise comparative outputs. More recently, simple SFT on large model outputs (also called *distillation*) has proven a pow-

erful tool enabling reasoning models (DeepSeek-AI et al., 2025). Unfortunately, the open source ecosystem supporting post-training in general, and data curation in particular, is in its infancy, with industry labs' capabilities far outstripping that of most academic labs (Weber et al., 2024; Feuer et al., 2024b; Ivison et al., 2023).

A stark challenge for smaller labs, especially in academia, has been the difficulty of acquiring publicly available synthetic datasets at large scale. This has posed barriers for researchers who are interested in conducting careful comparative analyses of the synthetic data quality (SDQ) of data generating models (DGMs), as measured by standard academic ground-truth and LLM-as-a-judge benchmarks.

To close this gap and better understand the downstream effects of DGM choice on synthetic data quality, we develop WILDCHAT-50M, which is the largest and most diverse publicly available dataset of chat transcripts to date. We also show that WILDCHAT-50M is a particularly effective source for post-training data for LLMs. Our core contributions in this work are as follows:

1. We introduce WILDCHAT-50M, the largest publicly available dataset of chat transcripts. Our dataset consists a vast corpus of synthetically generated chat transcripts using 50 different open-weight models. ranging in size from 0.5B to 104B parameters, each participating in over 1M multi-turn conversations. Each model participates in 2 or 3 turns per conversation on average, resulting in approximately a dataset comprising over 125 million chat transcripts in aggregate.

2. We conduct a thorough comparative analysis on the runtime and VRAM efficiency of these models, as well as analyze the distinctive qualities of different model outputs. Our analysis may inform researchers on how to scale up post-training data even further in the future.

3. We demonstrate the power of our dataset by using it as the basis of RE-WILD, a novel data mix for supervised fine-tuning (SFT) of LLMs. When we fine-tune Llama-3.1 8B Base on RE-WILD, we show that our models outperform the SFT mix proposed in Tulu-3 (Lambert et al., 2024), along with several other existing, strong SFT baselines on a range of post-training benchmarks.

---

[1]Department of Computer Science and Engineering, New York University, New York City, USA. Correspondence to: Benjamin Feuer <bf996@nyu.edu>.

*Proceedings of the 42nd International Conference on Machine Learning*, Vancouver, Canada. PMLR 267, 2025. Copyright 2025 by the author(s).

The rest of this paper is organized as follows. Section 2.1 describes high-level details of how the WILDCHAT-50M dataset was constructed. Section 2.2 provides a deep dive into the generation process of this dataset, along with technical aspects such as model response similarity and throughput efficiency. Section 3 lists and analyzes the results of our SFT experiments. Section 4 overviews related work, and Section 6 gives a concluding discussion.

## 2. The WILDCHAT-50M Dataset

### 2.1. Data Collection

We begin with a brief description of the data collection process for WILDCHAT-50M, detailing our source of prompts and the technical details of how responses were collected.

Although, to our knowledge, there are no large-scale diverse chat transcripts dataset of synthetically generated LLM responses, there are at least two recent large chat datasets available on which one could potentially base such a dataset: WildChat-1M from AllenAI, and LMSys-Chat-1M from LMSys (Zhao et al., 2024b; Zheng et al., 2024). Although both have their strengths, we chose to focus on the former because of its rich variety of use-cases (including those which are potentially toxic), its diverse regional and temporal dimensions, and its relatively low levels of contamination for commonly used test sets (Zhao et al., 2024b; Lambert et al., 2024).

**Technical details.** Our data collection process was conducted over a period of approximately two months on a 12x8 H100 shared research cluster. We estimate the total GPU costs of our data collection at 10,000 H100-hours. The homogeneity of the nodes in our study and codebase allow us to make controlled comparisons on important considerations, such as the VRAM efficiency and runtime of each model.

All responses and judgments are generated using VLLM (Kwon et al., 2023), a highly performant and stable framework for LLM inference. Models are distributed across up to 8 GPUs; we do not conduct infererence using more than one node for any model. We first minimize the number of GPUs required per model, and then heuristically maximize the size of the context window given that number of GPUs and the capacity of the model, resulting in a wide range of context windows, depending on the model architecture (2048 tokens to 20,000 tokens).

The largest models in our data collection process were queried using FP8 quantization, with checkpoints provided by Neural Magic (Kurtic et al., 2023). All other models were run in bfloat16, using their native checkpoints. We do not further ablate the effect of this quantization on output quality, as this has been studied and reported in prior work (Jin et al., 2024).

### 2.2. Dataset Analysis

WILDCHAT-50M collects data from 19 unique pre-trained models (each of which is post-trained) and 35 post-trained model variants (with non-unique pre-trained models). This yields a total of 54 DGMs represented. With the exception of the responses in the original WildChat dataset, which are sourced from various GPT checkpoints, all responses in WILDCHAT-50M are derived from models sourced from HuggingFace; the release dates range from July 2023 to November 2024, and the parameter counts range from 0.5B to 104B. For a comprehensive list of all the LLMs we used in the study, please refer to Sec. C. We attempted to select a diverse set of models; our main limiting factor was compatibility with our hardware setup, and with VLLM as an inference engine. The resulting public artifact is more than 50 times larger than the next largest public chat datasets of which we are aware, WildChat-1M and LMSys-Chat-1M.

**Naming Conventions.** In order to make this paper more readable, we will employ certain naming conventions for the models and datasets generated using DGMs described in this paper. The aforementioned conventions are enumerated below.

- We will sometimes utilize abbreviations for some particularly common model names: Qwen2.5-72B-Instruct := Q72, Llama-3.1-8B-Instruct := L8I, Llama-3.3-70B := L70, Qwen2-7B-Instruct := Q7, Cohere-Command-R-Plus-104B := CRP, AI21-Jamba-Mini-1.5-52B := JMB.
- Our model names will follow the following general convention: {SFT target : DGM}.
- Sometimes we will not specify the SFT target model name; in that case, it will always be Llama-3.1-8B-Base := L8B.
- Several times, we just report benchmarks for a model as is and not do any model post-training; in this case, the naming convention is just { Model name }: None.
- Most of our experiments were conducted on SFT models trained on 250,000 (250k) conversations. If we used a quantity other than 250k, we note it in the model name.

**Analysis of Throughput Efficiency.** Our first comparison describes the relative efficiency of inference across the models in our study. We consider two measures of throughput efficiency; average combined input and output tokens per second (**Tok/s**), and average time elapsed in seconds per 1000 conversations processed (**Time**). We compute our averages over a random subset of 5000 conversations.

The slowest model in our study is Qwen2.5-72B-Instruct with a context window length of 20,000, averaging 3,163 Tok/s, and the fastest is Llama-2-7B-Chat, with a context window of 2,048, averaging 37,357 Tok/s, more than

10 times faster. Input is significantly faster than output; the mean ratio over all unique pretrained models is 4.68 to 1, with a large standard deviation of 3.3. Both Time ($\sigma = 0.90, \rho = 0.73$) and Tok/s ($\sigma = -0.41, \rho = -0.80$) are strongly correlated with a simple proxy for model efficiency, the product of context window length and number of parameters.

**Analysis of Response Similarity.** To the best of our knowledge, the degree of similarity between diverse human respondents to LLM chat prompts has not been rigorously quantified at scale. However, this problem has been studied in the domain of abstractive summarization, where it can be assumed that the similarity would be considerably higher (Maynez et al., 2020; Iskender et al., 2021; Lin & Hovy, 2002; van Halteren & Teufel, 2003; Jing et al., 1998). For summarization tasks, there is no "one truth", evidenced by a low agreement between humans in producing gold standard summaries by sentence selection, low overlap measures between humans when gold standard summaries are created by reformulation in the summarizers' own words, and assigning information overlap between them.

It would be natural to assume, therefore, that LLMs that do not entirely share either pre-training or post-training data would likewise produce substantively different responses to prompts. However, our results show that this does not appear to be the case; LLM responses are unusually similar to one another. See Sec. A for a deeper analysis of this result, as well as every score (with associated standard deviation) for every model.

## 3. SFT Experiments

We now show that WILDCHAT-50M can be leveraged by researchers as a very valuable dataset for studying data curation strategies for LLMs post-training. Our core experiments focus on the SFT stage of post-training (also referred to as instruction tuning). While other forms of post-training (such as tuning to human preferences) are also interesting, we leave their thorough analysis to future work.

Following recent work such as Lambert et al. (2024), we focus on curating an SFT data mixture using a human-in-the-loop process, in contrast with an automated curation process such as that of Xu et al. (2023a). Unlike both of those works, we do not curate new prompts; only new responses to them.

Our dataset, samples and code are available at `https://github.com/penfever/wildchat-50m`.

### 3.1. RE-WILD: A new data mixture for SFT

Following the recent work of Lambert et al. (2024), we design our SFT data mixture, that we call RE-WILD, using a combination of WildChat data with a particularly high

| Source | Num. Convs |
|---|---|
| WildChat-Q72 | 246,750 |
| MMLU Auxiliary Train | 99,800 |
| Tulu 3 Persona Hub Algebra | 20,000 |

Table 1. **Data blending in RE-WILD.** *Our data blend is simpler than Tulu 3, consisting of just three sources, and is around 40% the size of the Tulu 3 SFT blend. The datasets were chosen heuristically to emphasize complementary skillsets (math, world-knowledge, and chat/instruction following). MMLU Auxiliary Train data is from Hendrycks et al. (2021), Tulu 3 Persona Hub Algebra is from Lambert et al. (2024).*

quality DGM that generates the responses and datasets designed to boost performance on world knowledge benchmarks. Later in this section, we describe the empirical process by which we determined which DGMs had high SDQ, and how.

The specific composition of our mix can be found in Tab. 1;. The datasets in this composition were chosen heuristically to emphasize complementary skillsets (math, world-knowledge, and chat/instruction following).

**Training.** We conduct our SFT experiments using a modified version of the Axolotl framework (Lian, 2025). We use the AdamW optimizer (Loshchilov & Hutter, 2017) with a learning rate of 2e-5, a single epoch, and a cosine learning rate scheduler, with eight steps of gradient accumulation, in bf16 precision. We also utilize several techniques to optimize training speed, such as gradient checkpointing, flash attention, and in some cases, FSDP (full shard, autowrap). The base model trained is always llama 3.1 8B (for us and baselines). All artifacts are available on our GitHub repo. Each of our SFT runs utilizes one 4xH100 node. The average time to fine-tune a model for 250,000 conversations takes approximately 5.5 hours.

**Evaluation.** Benchmarking LLM alignment is a challenging task, because of the open-ended nature of the objective and the large number of potential confounds (Lambert et al., 2024), as well as the fact that evaluation hyperparameters are not generally standardized across reported results. To deal with the latter issue, we employ Evalchemy, a recently introduced evaluation framework that is standardized, popular, reliable, and validated by reproduction reports for all benchmarks (Guha et al., 2024). Evalchemy itself utilizes the LM Evaluation Harness from Eleuther AI (Gao et al., 2024).

In order to make comparisons with past and future work easier, we select benchmarks which are popular and prominent in the recent research literature. We break down the concept of alignment into subcategories such as generalist chat capability, world knowledge, and instruction follow-

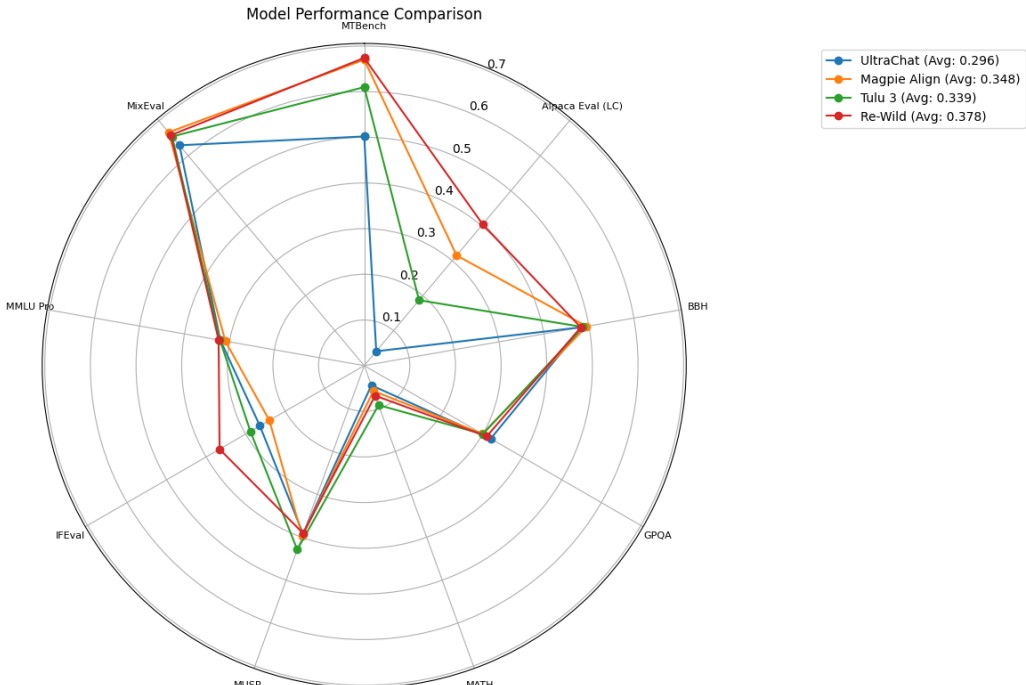

Figure 1. RE-WILD *outperforms strong baselines, on average, across nine benchmarks.* In particular, it exhibits strong performance on generalist chat and instruction following benchmarks. MT Bench scores here are divided by 10, so that the scale is similar to our other evaluations. For the exact numeric scores for all models, please refer to our GitHub repository. Figure best viewed in color.

| Model | MTBench | AlpacaEval | BBH | GPQA | MATH | MUSR | IFEval | MMLU Pro | MixEval |
|-------|---------|------------|-----|------|------|------|--------|----------|---------|
| Q72 | **6.86** | **41.00** | 0.48 ± 0.01 | 0.29 ± 0.04 | **0.06 ± 0.00** | 0.40 ± 0.04 | 0.37 ± 0.04 | 0.30 ± 0.01 | **0.65 ± 0.01** |
| L8I | 6.26 | 21.12 | 0.46 ± 0.01 | 0.30 ± 0.04 | 0.04 ± 0.00 | 0.37 ± 0.03 | **0.38 ± 0.04** | **0.33 ± 0.01** | **0.65 ± 0.01** |
| L70 | 6.23 | 24.91 | 0.47 ± 0.01 | 0.30 ± 0.04 | 0.04 ± 0.00 | 0.39 ± 0.04 | 0.34 ± 0.04 | 0.31 ± 0.01 | **0.65 ± 0.01** |
| Q7 | 6.03 | 17.26 | **0.49 ± 0.01** | 0.30 ± 0.04 | 0.03 ± 0.00 | **0.42 ± 0.04** | 0.29 ± 0.04 | 0.30 ± 0.01 | 0.61 ± 0.02 |
| CRP | 6.05 | 13.44 | **0.49 ± 0.01** | **0.31 ± 0.04** | 0.04 ± 0.00 | 0.39 ± 0.04 | 0.28 ± 0.04 | 0.32 ± 0.01 | 0.60 ± 0.02 |
| JMB | 6.05 | 25.14 | 0.47 ± 0.01 | 0.28 ± 0.04 | 0.04 ± 0.00 | 0.38 ± 0.04 | 0.26 ± 0.04 | 0.29 ± 0.01 | 0.57 ± 0.02 |

Table 2. **The choice of data generating model has strong and unpredictable effects on downstream benchmark performance.** We compare the performance of six different DGMs from four different model families, ranging in size from 0.5B to 104B parameters, each fine-tuned on 250k samples from a DGM. We find a large degree of variance in benchmark performance, with no one model dominating.

ing. Following recent work, we select a mix of ground-truth benchmarks and LLM-judge benchmarks, in order to balance out the potential confounds inherent to each evaluation method (Feuer et al., 2024a; White et al., 2024). For generalist chat capabilities, we use MixEval, AlpacaEval2, and MTBench with GPT-4o-mini as the judge LLM (Ni et al., 2024; Dubois et al., 2024; Zheng et al., 2023). In some cases, we also report the average score over all of our benchmarks. For AlpacaEval2, we report length-controlled win rate. For instruction following and world knowledge, we utilize the recent HuggingFace OpenLLM Leaderboard 2 (Fourrier et al., 2024). For IFEval, we report prompt-level strict accuracy because it is more challenging and therefore better exhibits separation between models; however, we also include instance-level loose accuracy in our artifacts, where we observe generally similar trends. Finally, in some figures and tables, we report the average performance for all benchmarks (MixEval, AlpacaEval2-LC, MTBench / 10, OpenLLM LB 2) as **Avg**. Where possible, we include 95% confidence intervals using the normal approximation method.

**Baselines.** Following (Lambert et al., 2024), we utilize strong baseline checkpoints also utilized in that paper; Tulu 3 SFT, Magpie Align SFT from Xu et al. (2024), and Ultrachat from (Cui et al., 2023).

### 3.2. Key Findings

**RE-WILD is a strong SFT data mix.** Our first result is that RE-WILD constitutes a particularly attactive data mixture for SFT. In the spider chart shown in Fig. 1, we show that RE-WILD outperforms several strong SFT baselines on aggregate; in particular, it excels on generalist chat tasks as well as instruction following tasks. Because prior work has shown that LLM judges can introduce implicit biases into their judgments, we include a mix of ground truth and LLM judged benchmarks (Feuer et al., 2024a). We find that RE-WILD performs well on both measures, indicating robust support for the claim that RE-WILD provides a superior model for generalist chat and instruction following.

**How much data from WILDCHAT-50M should be used?** Scaling up dataset size is a generally acceptable method for improving SFT model performance in most settings. But what sort of scaling laws apply for partially synthetic datasets such as WILDCHAT-50M? We ablate the effect of data scaling at 100k, 250k and 500k samples across four DGMs. In Fig. 2, we see that average performance steadily improves with scale, as expected, for most models. In this figure, GPT is used to denote the original WildChat dataset, which were generated using a blend of different GPT checkpoints, primarily 3.5. The upper asymptote for performance (if it exists) is beyond the maximum we have encountered in our experiments.

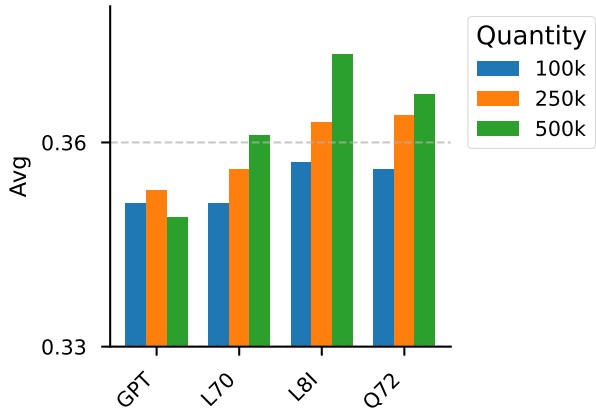

*Figure 2.* **Data scaling improves SFT performance.** *The effect is, however, somewhat dependent on SDQ – for DGMs such as GPT 3.5, the benefits taper off relatively quickly, but for the other three DGMs we consider, they continue to increase. **Avg** is the average performance over (MixEval, AlpacaEval2-LC, MTBench / 10, OpenLLM LB 2).*

**How much does the choice of data generating model impact downstream performance?** Can we be certain that it is not the prompts, or perhaps some quirk of our training procedure, that have led to improved SFT performance using RE-WILD? To evaluate this concern, we compare the performance of six unique pretrained models from four distinct model families, including Qwen-2.5-72B-Instruct from Alibaba, Llama-3.3-70B-Instruct from Meta, Command-R-Plus from Cohere, and Jamba-1.5-Mini from AI21 (Qwen Team, 2024; Dubey et al., 2024; Gomez; Lieber et al., 2024). The models range from 7B to 104B active parameters. The choice of DGM has a large effect, even when controlling for potential confounds such as the number of parameters for the model and the size of the context window. Furthermore, we find that no model dominates the benchmark, and that parameter count is not a perfect indicator of data quality. On three of the nine benchmarks we consider, the best performing model has fewer than 10B parameters. See Tab. 2.

**How much does the length of the context window impact performance?** Qwen-2.5-72B supports a large context window of over 131K tokens, which we were able to take advantage of during data generation. We experiment with truncating all Qwen-2.5-72B responses so that they are no longer than Llama-3.3-70B responses (with a context window of 8,192 tokens compared to 20,000 for Qwen in our experiments). Surprisingly, we find that the effect on the SFT model is slightly positive (**.404** vs **.400** averaged over 9 benchmarks). This is perhaps because we use a context window of 8,192 tokens in our SFT base model, Llama-3.1-8B.

**Do models benefit from blending DGMs?** One impetus behind creating a very large prompt-response dataset like that of Zhao et al. (2024b) is the intuition that more samples and more interactions will generally lead to better models. In the case of prompt diversity, this appears to be true (Feuer et al., 2024a), perhaps because it makes the model more robust to inconsistencies in prompting. But do the benefits of heterogeneous scaling extend to *responses*? In other words, do models trained on *blended* DGM responses outperform the sum of their parts? We conduct experiments to test this hypothesis, the results of which can be found in Sec. D, summarized here. We find that blending offers no benefit; the whole behaves almost exactly as the sum of its parts. This finding indicates that the dependencies on prompt diversity discovered in prior work do not extend to the space of model responses even for a large set of benchmarks. It is therefore most effective to optimize, rather than generalize, responses.

**Are models with strong performance on certain benchmarks better teachers for those benchmarks?** We evaluate this question by comparing a Llama-3.18B-Base model that is fine-tuned on Qwen-2.5-72B responses, to a Llama-3.1 8B-Base model fine-tuned on Llama-3.3-70B responses, and then comparing the two source models directly. We report the **agreement rate** between model checkpoints; we say that there is agreement if the fine-tuned model and the base model are both better or both worse than their counterparts, and there is not agreement otherwise. Across six benchmarks, we report an agreement rate of .5, which is at chance level, indicating that fine-tuned models do not necessarily inherit the strengths and weaknesses of their synthetic data generators.

**Is styling inherited during SFT?** Effective use of presentational styling elements such as HTML tags, attributes and inline properties can have a strong effect on a text's readability and clarity. We investigate whether such formal styling behaviors are inherited from the DGM during the SFT process. We select a subset of 80 turns from MTBench and examine the behavior of two DGMs (Qwen 2.5 72B and Llama 3.3 70B) and their finetunes. We convert the model responses from Markdown (which they commonly use in their responses) to HTML and report the **absolute and proportional frequency** of each styling tag. Absolute frequency is the raw count for each feature (see Tab. 3 rows 1:4). Proportional frequency, for rows $A$, $B$, and feature $F$, is given by $A[F] \div B[F]$. The closer this quantity is to 1, the more similar the model responses.

We report the raw results from this experiment in Tab. 3. Overall, we observe that SFTs track the styling of their DGMs very closely indeed; across all style features, the mean proportional frequency (MPF) of Qwen SFT compared to Qwen-2.5-72B is .91 across all features, and for

Llama, this score is 1.05. When we compare the SFTs to each other, by contrast, it is 2.61, indicating that the responses are much less similar.

**Do models learn better from DGMs in the same model family?** Recent work from Tajwar et al. (2024) has shown that approaches that use on-policy sampling or attempt to push down the likelihood on certain responses (i.e., employ a "negative gradient") outperform offline and maximum likelihood objectives. In this section, we inquire whether this finding extends not only to direct on-policy sampling, but something we might call *approximate on-policy sampling*, where a pretrained base model is fine-tuned on its own post-trained outputs, or those of a model from a similar family. Indeed, when we experiment with Qwen-2-7B and Llama-3-8B, we find that this approach produces stronger benchmark results when controlling for dataset size, and appears also to extend to a larger post-trained checkpoint in the same model family; see Sec. B for the complete results. We acknowledge that our experiments here are limited in scope, and consider this an important area of future study.

### 3.3. Why do some models outperform others as sources of synthetic data?

We showed above that the choice of DGM can have a dramatic effect on the performance of downstream LLMs fine-tuned on their responses. We now explore possible explanations for *why* SDQ varies so dramatically, even between superficially similar models.

We begin by eliminating some of the more obvious explanations. For example, we showed above that model parameter count and response length is not reliably predictive of performance, so these are likely not primary factors contributing to data quality.

Inspection of the benchmark results in Tab. 2 shows a greater variance in performance on chat-quality and instruction following benchmarks (such as IFEval, MixEval, MTBench, and AlpacaEval). A natural explanation for this observation would be that SDQ is domain-specific, and is inherited from the DGM. If this was the case however, then benchmark performance of fine-tunes would generally agree with those of the DGMs on those benchmarks. In reality, benchmark performance is not reliably inherited from the DGM. On the AlpacaEval leaderboard, Qwen 2.5-72B and Llama 3.3-70B are essentially tied; but Qwen 2.5-72B is a superior DGM (as measured by AlpacaEval performance) (Dubois et al., 2024).

A potential resolution to this phenomenon lies in noting that LLM judges utilize a range of judgment criteria, both explicit and implicit (Feuer et al., 2024a). Therefore, it is possible that supervised fine-tuning can significantly improve a model's (generalist) benchmark score by improving

| Model | Strong | Em | Ol | Ul | h1 | h2 | h3 | h4 | p | len |
|---|---|---|---|---|---|---|---|---|---|---|
| Q72 : None | 741 | 53 | 83 | 230 | 24 | 0 | 135 | 26 | 1217 | 6492 |
| L70 : None | 406 | 38 | 76 | 116 | 30 | 38 | 22 | 4 | 1222 | 6411 |
| L8B : L70 | 331 | 34 | 75 | 109 | 27 | 13 | 32 | 3 | 1210 | 6207 |
| L8B : Q72 | 808 | 60 | 77 | 260 | 20 | 0 | 149 | 33 | 1237 | 6585 |
| PF ∥ L8B : L70, L70 : None | 0.815 | 0.895 | 0.987 | 0.940 | 0.900 | 0.342 | 1.455 | 0.750 | 0.990 | 0.968 |
| PF ∥ L8B : Q72, Q72 : None | 1.090 | 1.132 | 0.928 | 1.130 | 0.833 | 1.000 | 1.104 | 1.269 | 1.016 | 1.014 |
| PF ∥ L8B : Q72, L8B : L70 | 2.441 | 1.765 | 1.027 | 2.385 | 0.741 | 0.000 | 4.656 | 11.000 | 1.022 | 1.061 |

*Table 3.* **SFT models strongly inherit formal stylistic elements from their DGMs.** *This table indicates how frequently certain Markdown stylistic elements appear in LLM responses (converted to HTML tags for greater clarity). The columns are names of HTML tags and inline properties, and the cells are frequency counts. PF stands for* **proportional frequency**, *the ratio of the first and second model listed (order here is presumed to be arbitrary).*

on only one or two criteria (e.g., comprehensiveness and readability), while its DGM, whose overall score may be already higher, comparatively underperforms because of limitations in some other criteria such as factuality.

In order to determine whether this is indeed the case in our evaluations, we conduct an experiment using 80 turns from MT Bench drawn from four models — (i) Q72 : None, (ii) L70 : None (DGMs), (iii) L8B : Q72, and (iv) L8B : L70 — each fine-tuned on 500k samples from WILDCHAT-50M. We provide the complete conversations in the appendix for reference.

In Tab. 4, we provide raw results in the form of MTBench score as well as PrefRt, which stands for **Preference Rate**, or the rate at which a model's responses are preferred over another's (100 means always preferred, 0 means never preferred). L8L is the rate at which a model's response is preferred over L8B : L70, L8Q is the rate at which a model's response is preferred over L8B : Q72, L70B is the rate at which a model's response is preferred over L70: None, and Q72B is the rate for Q72 : None.

We can see that L8B : Q72 outperforms L8B : L70, and Q72 : None outperforms L70 : None, both in terms of average MTBench score, and in terms of win rate. Interestingly, the proportional improvement is similar in both comparisons, suggesting a degree of heritability to LLM judge preferences. When we manually inspected a random sample of model outputs, we often agreed with the judge (the reader may form their own opinions by referring to examples in Sec. E). The MT bench prompts, responses and complete judgments are available in our GitHub repository.

When we include the DGMs themselves in the comparisons, the conclusions are somewhat different. L70 : None outperforms L8B : Q72 on both metrics. In particular, on 5 of the 80 turns, L70 : None flips the ranking (rather than simply breaking a tie), compared to L8B : L70. We note first that such judgment flips are quite rare. We manually inspect these 5 flips and find that on 4 of the 5, the LLM judge cited *factuality* as a key reason for L8B : L70's low

score. In other words, even though the style of L8B : Q72 is still generally preferred by the judge, the superior factuality of L70 : None drives an overall change in rank.

We analyze the same phenomenon through another lens in Fig. 3 and Fig. 4). Here, we visualize the frequency of common words in the judgments where the score differed between models. Common negations of words were counted separately (although these were rare). We discover that L8B : L70 responses are more commonly associated with words such as "lacks", "few", "misleading" and "concise", whereas L8B : Q72 responses are more commonly associated with words such as "appropriate", "comprehensive", "complete" and "detailed".

One limitation of this analysis is that it fails to capture common context that tends to accompany these words in the judgments; for example, the reason that "clearer" is more common in L8B : L70 judgments is that the judge frequently employed semantic variations of phrases like "could have been clearer", not because the responses themselves were "clearer". The same applies to "critical".

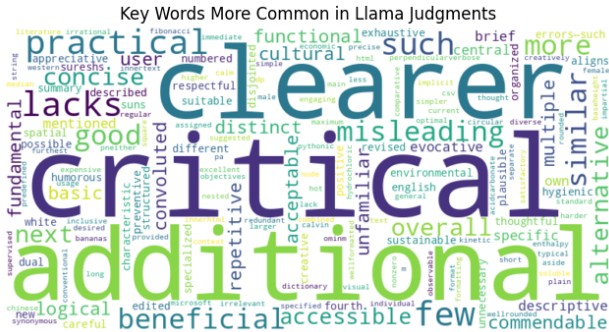

*Figure 3.* **Key words more common in L8B : L70 judgments.** *The more negative tone of these judgments emphasizes words like clearer (as in, "could have been clearer"), lacks, convoluted and repetitive.*

From these results, it seems that SDQ on a set of generalist

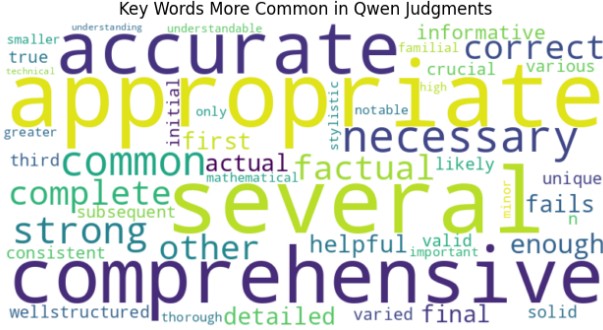

Figure 4. **Key words more common in L8B : Q72 judgments.** *These judgments tended to be more positive; emphasis was placed on words like appropriate, necessary, comprehensive and accurate.*

| Model | MTBench | PrefRt-L8L | PrefRt-L8Q | PrefRt-L70B | PrefRt-Q72B | FlipCt |
|---|---|---|---|---|---|---|
| L8B : L70 | 6.38 | N/A | 22.5 | 8.75 | 15 | N/A |
| L8B : Q72 | 6.72 | 40 | N/A | 16.25 | 13.75 | N/A |
| L70 : None | 7.64 | 52.5 | 38.75 | N/A | 21.25 | 5 |
| Q72 : None | 7.83 | 61.25 | 43.75 | 25 | N/A | 6 |

Table 4. **LLM judge preferences for DGMs and SFTs.** *For column name definitions, we refer the reader to the main text, section Sec. 3. Overall, we observe that Qwen responses are generally preferred by LLM judges, and that the rate at which they are preferred is similar from DGM to SFT. Last but not least, we note that reversals of judgment from SFT to DGM (FlipCt) are uncommon.*

chat prompts such as WildChat is largely a function of domain-agnostic factors, such as the *comprehensiveness* of the model response, the *clarity* of the structure, and the *tone* and stylistic tendencies of the language.

## 4. Related Work

Our work explores LLM post-training, a research area which has witnessed dynamic growth since 2022. SFT, sometimes referred to as *instruction tuning* (Mishra et al., 2022; Wei et al., 2022; Sanh et al., 2022; Wang et al., 2022; Longpre et al., 2023), in which language models are trained on samples including task instructions and their corresponding responses, has been shown to allow LLMs to generalize better to unseen tasks. Initially those samples were drawn from traditional NLP tasks with verified ground truth answers (Wang et al., 2023b). However, over time, it has become clear that a more heterogeneous approach, both to prompt and model responses, tends to lead to superior outcomes, and that combining instruction tuning datasets, either strategically or randomly, can lead to strong results (Taori et al., 2023; Conover et al., 2023; Wang et al., 2023a).

Unfortunately, post-training is an area where open science continues to trail closed advances made in frontier indus-

try labs (Chiang et al., 2024). Models trained on open data underperform those trained on closed data, both in the pretraining and in the post-training stage. Numerous algorithmic advances have been introduced into the recent literature which attempt to serve as scalable instruction tuning methods (Tunstall et al., 2023; Xu et al., 2023b; Zhou et al., 2023; Yasunaga et al., 2024).

Despite the variety (and abundance) of algorithmic approaches, many fail to scale to the high-data regime (Feuer et al., 2024a). Therefore, in this work we turn to the more basic question of the quality of datasets used in post-training. While there is still considerable ground to be covered, we hope that the size and diversity of our dataset, combined with its strong performance on SFT benchmarks, will encourage researchers to use it as a basis for future work.

Recently, Zhao et al. (2024a) introduced WildChat-1M, a dataset containing 1 million ChatGPT interaction logs collected in real-world settings. Their work takes a different approach to ours by focusing on capturing authentic human-AI interactions rather than generating synthetic conversations. WildChat-1M leverages multiple GPT model variants, with 76% of conversations using the GPT-3.5-Turbo API (including versions 3.5-turbo-0613, 3.5-turbo-0301, and 3.5-turbo-0125) and 24% utilizing the GPT-4 API (including 4-1106-preview, 4-0314, and 4-0125-preview). For human prompt acquisition, they deployed chatbot services on Hugging Face Spaces and implemented a two-step consent process, collecting anonymized data from users who explicitly opted into the research. While WildChat-1M offers valuable insights into real-world chatbot interactions across multiple languages, our approach with WildChat-50m explores the complementary direction of high-volume synthetic data for post-training.

## 5. Limitations

There are a few key limitations of this work which we wish to highlight here. Because of practical constraints, we were not able to report results using other post-training approaches other than SFT. It is likely that the relative effect of DGM choice would differ depending on the post-training regime. Although our benchmark suite is standardized, balanced and large, it does not encompass all use cases. In particular, we did not evaluate performance on highly specialized tasks, such as coding or legal reasoning. We consider both of these limitations as pointing towards useful directions for future work. It might also be beneficial to consider the differential advantages and disadvantages of diversifying DGMs on smaller, more focused datasets, with and without ground truth.

# 6. Conclusions

In this work, we make several valuable practical and empirical contributions to the field's understanding of synthetic data. In particular, we provide robust empirical evidence that the choice of DGM is an extremely important factor in downstream SFT model performance on generalist chat benchmarks; simply by selecting a good DGM, we compensate for a small dataset size and outperform more complex methods and carefully curated SFT mixes.

Equally important, we provide novel insight into *why* certain DGMs produce much higher SDQ than others; our experiments indicate that *comprehensiveness, clarity, tone and prompt responsiveness* are highly heritable during the SFT process, even on generalist data, unlike skills such as world knowledge or mathematics, which are heritable only when data is curated for that particular purpose.

Finally, we provide novel comparative insights into LLMs, reporting a high degree of similarity in the prompt responses of diverse LLMs. Taken together with the prior observation, the distinction between high and low SDQ may be subtle, and worthy of future research.

## Impact Statement

This paper presents work whose goal is to advance the field of machine learning. There are many potential societal consequences of our work. In particular, we wish to acknowledge the fact that we are building upon a dataset (WildChat) which is known to contain examples of user inputs on potentially upsetting topics, including but not limited to sex, violence, and bigoted claims (Zhao et al., 2024b). Following the authors of that work, we will require approval before researchers are allowed to access our datasets, and will release our data under the AI2 ImpACT License, which explicitly forbids certain use cases for this data.

In addition, our data will be subject to any additional restrictions from the licenses of the models we utilize. We also preserve, without modification, the original user inputs from WildChat (Zhao et al., 2024b), and therefore in our work there remains the possibility that users may have inadvertently included personal information within their conversations which was not detected by the (considerable) safeguards put in place by the WildChat authors. We will make this clear to users upon dataset release.

## Acknowledgements

This work is supported in part by the AI Research Institutes program supported by NSF and USDA-NIFA under Award No. 2021-67021-35329, a Google Cyber NYC gift grant, and NSF grant # 2154119. The authors gratefully acknowledge the support of NYU IT's High Performance Computing resources, services, and staff. The authors also gratefully acknowledge compute support provided by the National Artificial Intelligence Research Resource (NAIRR) Pilot, Groq LPU Inference Engine, and the Empire AI Consortium.

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

| Model | avg_rouge1 | std_rouge1 | avg_rougeL | std_rougeL | avg_meteor | std_meteor |
|---|---|---|---|---|---|---|
| Mixtral-8x7B-Instruct | 0.37 | 0.11 | 0.19 | 0.06 | 0.20 | 0.05 |
| Llama-3.1-Nemotron-70B-Instruct | 0.37 | 0.06 | 0.23 | 0.05 | 0.20 | 0.05 |
| Qwen2.5-72B-Instruct | 0.34 | 0.09 | 0.17 | 0.05 | 0.17 | 0.05 |
| Mistral-7B-wizardlm | 0.34 | 0.07 | 0.22 | 0.06 | 0.16 | 0.05 |
| Mistral-7B-sharegpt-vicuna | 0.34 | 0.06 | 0.18 | 0.03 | 0.18 | 0.04 |
| Mistral-7B-Base-SFT-IPO | 0.33 | 0.11 | 0.17 | 0.05 | 0.19 | 0.06 |
| internlm2_5-20b-chat | 0.33 | 0.08 | 0.15 | 0.03 | 0.20 | 0.06 |
| Llama-3.1-70B-Instruct | 0.33 | 0.10 | 0.16 | 0.05 | 0.16 | 0.06 |
| Llama-3.3-70B-Instruct | 0.33 | 0.09 | 0.17 | 0.04 | 0.20 | 0.05 |
| Llama-2-7b-chat-hf | 0.32 | 0.09 | 0.19 | 0.07 | 0.20 | 0.06 |
| Mistral-7B-Base-SFT-CPO | 0.32 | 0.07 | 0.17 | 0.04 | 0.18 | 0.05 |
| Qwen2-7B-Instruct | 0.32 | 0.08 | 0.15 | 0.04 | 0.17 | 0.06 |
| Llama-3-8B-ShareGPT-112K | 0.31 | 0.09 | 0.18 | 0.07 | 0.15 | 0.06 |
| Qwen2.5-Coder-32B-Instruct | 0.31 | 0.10 | 0.15 | 0.05 | 0.18 | 0.05 |
| Llama-3-8B-Magpie-Pro-SFT-200K | 0.30 | 0.15 | 0.18 | 0.09 | 0.17 | 0.09 |
| google_gemma-2-9b-it | 0.30 | 0.10 | 0.16 | 0.06 | 0.14 | 0.06 |
| AI21-Jamba-1.5-Mini | 0.29 | 0.13 | 0.17 | 0.08 | 0.14 | 0.06 |
| OpenHermes-2-Mistral-7B | 0.28 | 0.09 | 0.16 | 0.06 | 0.15 | 0.07 |
| Llama-3-Base-8B-SFT-ORPO | 0.27 | 0.07 | 0.14 | 0.03 | 0.22 | 0.05 |
| google_gemma-2-27b-it | 0.27 | 0.04 | 0.13 | 0.02 | 0.15 | 0.03 |
| OpenHermes-2.5-Mistral-7B | 0.27 | 0.06 | 0.15 | 0.04 | 0.13 | 0.04 |
| Mistral-7B-Base-SFT-SLiC-HF | 0.26 | 0.13 | 0.14 | 0.07 | 0.13 | 0.07 |
| Mistral-7B-Base-SFT-KTO | 0.25 | 0.05 | 0.13 | 0.04 | 0.11 | 0.04 |
| Ministral-8B-Instruct-2410 | 0.24 | 0.13 | 0.12 | 0.07 | 0.13 | 0.09 |
| Llama-3-Base-8B-SFT-RDPO | 0.23 | 0.06 | 0.13 | 0.03 | 0.19 | 0.03 |
| Mistral-7B-Base-SFT-RRHF | 0.19 | 0.06 | 0.08 | 0.02 | 0.16 | 0.04 |

*Table 5.* **Intra-LLM response similarity in** RE-WILD. *Here we report intra-llm response similarity scores. The method we use to obtain these scores is described in Sec. 2.2.*

## A. Intra-LLM Response Similarity

In our experiment, we collect 500 responses from 25 randomly selected models and compute their similarity to a set of reference responses (randomly sampled from 4 other models in the set of 25) using three traditional NLP similarity metrics: ROUGE-1, ROUGE-L, and METEOR (Banerjee & Lavie, 2005; Ganesan, 2018). These metrics are computed as F1-score over unigrams, F1-score using the longest common subsequence, and weighted F1-score giving 9:1 weightage for precision over recall with a chunking penalty. In Tab. 5, we include the extended results for response similarity. To see a relevant citation for any particular model in this table, please refer to Sec. C.

Overall, we find high similarity across models, albeit with a fairly high degree of variance; for example, Mixtral-8x7B-Instruct has a high ROUGE-1 similarity of 0.37, while Ministral-8B-Instruct-2410 has 0.24, among the lowest scores. Considering the diversity of both prompts and models, this level of similarity suggests that LLMs produce much more regular and predictable output than humans. On a model-by-model level, we can interpret measures like these as signal as to "how generic" and "how consensus-driven" any particular LLM's response is. We also observe that the larger models tend to generate *more* similar responses; in some sense, they are closer to a consensus response to the prompt.

## B. Ablation on the Effect of On-Policy DGMs

In Tab. 6, we provide extended results on the effects of fine-tuning using DGMs that are highly similar to the fine-tune targets (and therefore, in some limited sense, on-policy). For the analysis of the results here, please refer to our main paper, Sec. 3. The naming convention used in this table is described in Sec. 2.2, with one new abbreviation; Qwen-2-7B-Base := Q7B.

| Model | MTBench | AlpacaEval | BBH | GPQA | MATH | MUSR | IFEval | MMLU Pro | MixEval | Avg |
|---|---|---|---|---|---|---|---|---|---|---|
| L8B : L8I | 6.26 | 21.12 | 0.46 ± 0.01 | 0.30 ± 0.04 | 0.04 ± 0.00 | 0.37 ± 0.03 | 0.38 ± 0.04 | 0.33 ± 0.01 | 0.65 ± 0.01 | 0.36 |
| L8B : Q7 | 6.03 | 17.26 | 0.49 ± 0.01 | 0.30 ± 0.04 | 0.03 ± 0.00 | 0.42 ± 0.04 | 0.29 ± 0.04 | 0.30 ± 0.01 | 0.61 ± 0.02 | 0.35 |
| L8B : L70 | 6.23 | 24.91 | 0.47 ± 0.01 | 0.30 ± 0.04 | 0.04 ± 0.00 | 0.39 ± 0.03 | 0.34 ± 0.04 | 0.31 ± 0.01 | 0.65 ± 0.01 | 0.36 |
| Q7B : L8I | 6.51 | 15.87 | 0.51 ± 0.01 | 0.29 ± 0.04 | 0.17 ± 0.00 | 0.43 ± 0.04 | 0.35 ± 0.04 | 0.39 ± 0.01 | 0.61 ± 0.02 | 0.39 |
| Q7B : Q7 | 6.69 | 27.09 | 0.54 ± 0.01 | 0.31 ± 0.04 | 0.19 ± 0.01 | **0.45 ± 0.04** | 0.40 ± 0.04 | 0.42 ± 0.01 | 0.69 ± 0.01 | 0.43 |
| Q7B : Q72 | **7.25** | **36.68** | 0.54 ± 0.01 | 0.32 ± 0.04 | **0.21 ± 0.01** | 0.43 ± 0.04 | 0.34 ± 0.04 | **0.43 ± 0.01** | 0.65 ± 0.01 | 0.42 |
| Q7 : None | 7.17 | 33.14 | **0.55 ± 0.01** | **0.33 ± 0.04** | 0.19 ± 0.01 | **0.45 ± 0.04** | **0.42 ± 0.04** | 0.40 ± 0.01 | 0.73 ± 0.01 | **0.44** |
| L8I : None | 7.20 | 30.84 | 0.51 ± 0.01 | **0.33 ± 0.04** | 0.12 ± 0.01 | 0.40 ± 0.03 | **0.42 ± 0.04** | 0.38 ± 0.01 | **0.74 ± 0.01** | 0.41 |

*Table 6. **Models learn more effectively from highly similar DGMs.** In this table, we report the complete, extended results from our experiments on the effect of diversifying both DGM and SFT-target. Both Llama and Qwen benefit from more similar upstream models.*

## C. List of All LLMs in the Study, with Citations

We attempt to use model naming conventions consistent with those on HuggingFace; that way, in order to find any particular model, it should only be necessary to Google its name. We do not include the complete HuggingFace links because they might de-anonymize this work. Where available, we include a citation to the work where the model was first introduced into the literature.

- NVLM-D-72B, from Dai et al. (2024)

- Llama-3.3-70B-Instruct, Llama-3.1-8B-Instruct, Llama-3.1-70B-Instruct from Dubey et al. (2024)

- Yi-1.5-34B-Chat, from AI et al. (2025)

- c4ai-command-r-plus-08-2024, from Gomez

- mistral-7b-sft-beta, from Tunstall et al. (2023)

- Llama-3-8B-Magpie-Align-v0.2, Llama-3-8B-Magpie-Pro-SFT-200K-v0.1, Llama-3-8B-OpenHermes-243K, Llama-3-8B-ShareGPT-112K, Llama-3-8B-Tulu-330K, Llama-3-8B-Ultrachat-200K, Llama-3-8B-WildChat, Llama-3-8B-WizardLM-196K, from Xu et al. (2024)

- Athene-70B, from NexusFlow AI (2025)

- Qwen2-7B-Instruct, Qwen2.5-14B-Instruct, Qwen2.5-72B-Instruct, Qwen2.5-Coder-32B-Instruct from Qwen Team (2024)

- glm-4-9b-chat from GLM et al. (2024)

- AI21-Jamba-1.5-Mini from Lieber et al. (2024)

- gemma-2-27b-it, gemma-2-9b-it from Team et al. (2024)

- internlm2_5-20b-chat from Cai et al. (2024)

- Llama-2-7b-chat-hf, Llama-2-13b-chat-hf from Touvron et al. (2023)

- Ministral-8B-Instruct-2410, Mistral-Nemo-Instruct-2407, Mixtral-8x7B-Instruct-v0.1 from Mistral (2024); Jiang et al. (2023)

- Llama-3.1-Nemotron-70B-Instruct-HF from Adler et al. (2024)

- Mistral-7B-magpie-v1.0, Mistral-7B-sharegpt-vicuna-v1.0, Mistral-7B-tulu, Mistral-7B-wizardlm-v1.0 from Feuer et al. (2024a)

- Llama-3-Base-8B-SFT-CPO, Llama-3-Base-8B-SFT-DPO, Llama-3-Base-8B-SFT-IPO, Llama-3-Base-8B-SFT-KTO, Llama-3-Base-8B-SFT-ORPO, Llama-3-Base-8B-SFT-RDPO, Llama-3-Base-8B-SFT-RRHF, Mistral-7B-Base-SFT-CPO, Mistral-7B-Base-SFT-DPO, Mistral-7B-Base-SFT-IPO, Mistral-7B-Base-SFT-KTO, Mistral-7B-Base-SFT-RDPO, Mistral-7B-Base-SFT-RRHF, Mistral-7B-Base-SFT-SLiC-HF, Mistral-7B-Base-SFT-SimPO from Meng et al. (2024)

- OpenHermes-2-Mistral-7B, OpenHermes-2.5-Mistral-7B from Teknium et al. (2024)

## D. Ablations on Blending Data-Generating Models

Our extended results on the effect of blending DGMs can be found at Tab. 7. The analysis of this table can be found in our main paper, as well as explanations of the abbreviation conventions for model names.

| Model | MTBench | Alpaca Eval (LC) | BBH | GPQA | MATH | MUSR | IFEval | MMLU Pro | MixEval | Avg |
|---|---|---|---|---|---|---|---|---|---|---|
| L8B : Q7 (500k) | 6.33 | 19.51 | 0.48 ± 0.01 | 0.28 ± 0.04 | 0.04 ± 0.00 | 0.40 ± 0.03 | 0.33 ± 0.04 | 0.30 ± 0.01 | 0.64 ± 0.01 | 0.35 |
| L8B : L8I (500k) | 6.52 | 21.03 | 0.46 ± 0.01 | 0.32 ± 0.04 | 0.05 ± 0.00 | 0.39 ± 0.03 | 0.42 ± 0.04 | 0.32 ± 0.01 | 0.66 ± 0.01 | 0.37 |
| L8B : L8I + Q7 (500k) | 6.43 | 18.57 | 0.47 ± 0.01 | 0.28 ± 0.04 | 0.05 ± 0.00 | 0.41 ± 0.04 | 0.34 ± 0.04 | 0.32 ± 0.01 | 0.64 ± 0.01 | 0.36 |
| L8B : Q72 (500k) | 6.51 | 41.67 | 0.48 ± 0.01 | 0.29 ± 0.04 | 0.05 ± 0.00 | 0.39 ± 0.03 | 0.39 ± 0.04 | 0.30 ± 0.01 | 0.66 ± 0.01 | 0.37 |
| L8B : L70 (500k) | 6.39 | 27.38 | 0.46 ± 0.01 | 0.31 ± 0.04 | 0.04 ± 0.00 | 0.36 ± 0.03 | 0.38 ± 0.04 | 0.32 ± 0.01 | 0.65 ± 0.01 | 0.36 |
| L8B : Q72 + L70 (500k) | 6.82 | 39.93 | 0.48 ± 0.01 | 0.29 ± 0.04 | 0.04 ± 0.00 | 0.38 ± 0.03 | 0.38 ± 0.04 | 0.31 ± 0.01 | 0.65 ± 0.01 | 0.36 |

Table 7. **Effects of blending DGMs.** *Where we blend models, we always draw a random sample of approximately equal size from each DGM, and denote the mixture as DGM A + DGM B. e.g., L8I + Q7. In this table, it can be seen that blends benchmark between their DGMs, not above them (as measured by Avg).*

## E. Output Samples

In this appendix section, we provide complete sample responses from a pair of DGMs in conveniently human-readable format, along with the prompt(s) in the conversation; Llama-3.3-70B and Qwen-2.5-72B-Instruct are the models shown here. For more samples, including sample judgments, please refer to our GitHub repository. The responses may be more convenient to read in the repository than they are in this document, as they are HTML formatted; however, we include them here so as not to inconvenience the reader.

Please note that some of the output samples may contain statements that are offensive, toxic or otherwise distasteful to some readers. Please exercise caution when reading both prompts and responses. The content of the prompts and responses does not necessarily reflect the beliefs of the authors.

Describe gas giant with gothic colors. Including chemistry composition.

## allenai_WildChat-1M-Full-meta-llama_Llama-3.3-70B-Instruct

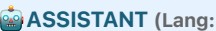**ASSISTANT** (Lang: English | Finish: stop)

The somber, mystical realm of the gas giant. Shrouded in an eerie, dark ombré,
like the mourning veil of a bereaved widow. The hue of the sky is a deep,
foreboding indigo, gradating into a sickly, yellowish-green towards the horizon,
as if tainted by the faint glow of luminescent decay.

The atmosphere itself is a noxious brew of hydrogen (H2) and helium (He), with
wispy tendrils of methane (CH4) and ammonia (NH3) wafting through the upper
strata, like the ghostly whispers of forgotten souls. These gases writhe and
twist in torment, forming swirling storm systems that churn with a maddening
fury, as if fueled by the very essence of despair.

As we delve deeper into the heart of the gas giant, the pressure and temperature
escalate, forging an environment that is at once both beautiful and terrifying.
The chemistry composition shifts, with hydrogen and helium

## allenai_WildChat-1M-Full-Qwen_Qwen2.5-72B-Instruct

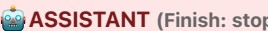**ASSISTANT** (Finish: stop)

Certainly! Imagine a gas giant that embodies the essence of gothic aesthetics, a
celestial body that is both majestic and foreboding, shrouded in deep, dark hues
and enveloped in a mysterious, almost sinister atmosphere.

### Appearance

The gas giant's atmosphere is a hauntingly beautiful tapestry of gothic colors.
The sky is dominated by deep purples, rich blacks, and dark blues, with
occasional streaks of blood-red and silver. The clouds, thick and swirling, are
tinged with a murky, almost metallic sheen. Lightning storms, rare but intense,
illuminate the atmosphere with flashes of electric blue and green, casting eerie
shadows across the gas giant's surface.

### Chemistry Composition

1. **Hydrogen (H$_2$)**: The primary component of the gas giant's atmosphere, hydrogen makes up approximately

succumbing to the
crushing force, and heavier elements such as carbon, nitrogen, and oxygen
beginning to assert their dominance. Water, ammonia, and methane ices form, like
the frozen tears of the damned, suspended in a sea of liquid hydrogen and
helium.

The planet's core, a glowing, hellish ember, pulsates with an otherworldly
energy, as if fueled by the very essence of darkness itself. The magnetic field
that surrounds it is a twisted, nightmarish latticework, crackling with
electrical energy, and weaving a spell of eerie, gothic power.

In this forsaken realm, the winds howl like a chorus of the doomed, while the
lightning flashes with an eldritch, greenish-blue glow, casting an eerie,
macabre light upon the churning clouds. The gas giant is a world of twisted,
gothic horror, a place where the very fabric of reality seems to be warped and
distorted, and where the darkness reigns supreme.

Chemical Composition:
- Hydrogen (H2): 75-80%
- Helium (He): 20-25%
- Methane (CH4): 1-5%
- Ammonia (NH3): 0.1-1%
- Water (H2O): 0.1-1%
- Carbon, nitrogen, and oxygen compounds: 0.01-0.1%

90% of the atmosphere by volume. It appears as a deep, almost black hue due to the presence of various compounds and the thick, absorbing nature of the upper atmosphere.

2. **Helium (He)**: The second most abundant element, helium comprises about 9%
of the atmosphere. It contributes to the overall density and opacity of the
atmosphere, enhancing the gothic, shadowy appearance.

3. **Methane ($CH_4$)**: Methane, present in lesser amounts, gives the planet its
deep blue and purple hues. The methane molecules absorb red light, leaving the
atmosphere with a predominantly blue and purple coloration. In the upper layers,
methane condenses into clouds, contributing to the gothic, stormy appearance.

4. **Ammonia ($NH_3$)**: Ammonia is present in the form of ice particles and
clouds, adding to the planet's dark, stormy appearance. These clouds are often
tinged with dark red and black, creating a moody and dramatic effect.

5. **Water ($H_2O$)**: Water vapor is also present, contributing to the planet's
atmospheric complexity. Water ice and vapor form high-altitude clouds that can
appear as ghostly, translucent layers, enhancing the gothic aesthetic.

6. **Sulfur Compounds**: Sulfur dioxide ($SO_2$) and hydrogen

Colors:
- Deep indigo (#3B3F54) to sickly yellowish-green (#8B9467)
- Faint, glowing blue-green (#34A8FF) of luminescent decay
- Hellish, ember-like glow of the core (#FF9900)
- Eerie, greenish-blue lightning flashes (#33CCFF)

This gas giant is a place of dark, gothic beauty, where the boundaries between
reality and nightmare are blurred, and the very essence of the universe seems to
be twisted and distorted.

sulfide ($H_2S$) are
minor but significant components. These compounds contribute to the planet's
reddish and brownish hues, creating a contrast that adds depth and forebroadness
to the atmosphere.

7. **Phosphine ($PH_3$)**: Phosphine is a trace gas that can create spectacular,
   bioluminescent clouds and lightning storms. During these storms, the phosphine
   reacts with other chemicals, producing flashes of electric blue and green,
   adding to the eerie, gothic ambiance.

8. **Metallic Hydrogen**: Deep within the planet, under immense pressure and
   temperature, hydrogen exists in a metallic state. This metallic hydrogen is
   responsible for the planet's strong magnetic field and contributes to its
   overall energy and dynamism.

## Additional Features

- **Rings**: The gas giant is encircled by dark, narrow rings composed of small, rocky debris and ice particles. These rings reflect the deep, moody colors of the planet, adding to the gothic atmosphere.
- **Moons**: The planet has several moons, each with its own gothic charm. Some moons are shrouded in dark, icy surfaces, while others have volcanic activity that produces plumes of sulfur and other volatiles, creating a stark, surreal landscape.

This gas giant, with its gothic colors and complex chemistry, is a mesmerizing

and haunting sight, a true marvel of the universe.

**Shared Prompt:**

[Response must be written as a short blurb.]
As of 2023, what is the copyright status of Oswald the Lucky Rabbit?

---

**allenai_WildChat-1M-Full-meta-llama_Llama-3.3-70B-Instruct**

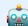 **ASSISTANT** (Lang: English | Finish: stop)

As of 2023, Oswald the Lucky Rabbit is now in the public domain in the United
States, having been created in 1927 and the original copyright expiring after 95
years.

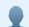 **USER** (Lang: English)

What is the different between a tax queen and welfare queen? Is tax queen a less
racially/ethnically pejorative term?

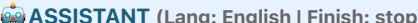 **ASSISTANT** (Lang: English | Finish: stop)

A "welfare queen" typically refers to a pejorative stereotype of a person, often
a woman of color, who allegedly abuses the welfare system by collecting
excessive benefits while not working. In contrast, a "tax queen" is not a
commonly used term, but it could potentially refer to someone who takes
advantage of tax loopholes or deductions to minimize their

---

**allenai_WildChat-1M-Full-Qwen_Qwen2.5-72B-Instruct**

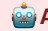 **ASSISTANT** (Finish: stop)

As of 2023, Oswald the Lucky Rabbit, created by Walt Disney
and Ub Iwerks, is in
the public domain. This means that anyone can use the
character without
infringing on copyright laws, providing a wide range of
creative opportunities
for artists and producers.

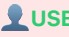 **USER**

What is the different between a tax queen and welfare
queen? Is tax queen a less
racially/ethnically pejorative term?

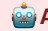 **ASSISTANT** (Finish: stop)

The terms "tax queen" and "welfare queen" are both used to
describe individuals
who allegedly manipulate the system to their advantage, but
they refer to
different aspects of public assistance and have different
connotations.

tax liability.

While "tax queen" may seem like a less racially or ethnically pejorative term,
it's essential to consider the context in which it's used. If the term is
applied primarily to individuals from marginalized groups, it could still
perpetuate negative stereotypes. However, if used to describe individuals of any
background who exploit tax laws, it might be seen as more neutral. Nonetheless,
the term "welfare queen" has a well-documented history of being used to
stigmatize and demean low-income communities, particularly communities of color,
making it a more loaded and problematic term.

👤**USER** (Lang: English)

Is it illegal in a British-style constitutional monarchy for a monarch to use
their royal status to avoid paying taxes?

🤖**ASSISTANT** (Lang: English | Finish: stop)

In a British-style constitutional monarchy, the monarch's tax status can be
complex. While the monarch is not above the law, they may be exempt from certain
tax obligations due to historical or customary arrangements. However, using
royal status to deliberately avoid paying taxes would be considered unethical
and potentially illegal. In the UK, for example, the Queen

**Welfare Queen:**
- **Definition:** A "welfare queen" is a derogatory term used to describe someone, often a woman, who is perceived to be gaming the welfare system to receive benefits fraudulently or excessively.
- **Connotations:** The term has a strong negative and often racially charged history, particularly in the United States, where it has been used to stereotype and stigmatize poor, often Black, women. It gained prominence in the 1970s and has been associated with racial and class biases.

**Tax Queen:**
- **Definition:** A "tax queen" is a more recent and less common term that typically refers to an individual, often a wealthy person, who uses legal loopholes and strategies to minimize or avoid paying taxes.
- **Connotations:** The term "tax queen" is generally not as racially or ethnically charged as "welfare queen." It is more likely to be used in financial or political contexts to criticize the exploitation of tax laws by the wealthy.

**Comparison:**
- **Racial and Ethnic Connotations:** "Welfare queen" is heavily laden with racial and ethnic stereotypes, particularly targeting minority groups. "Tax queen," while still derogatory, is less likely to carry the same racial overtones and is more focused on the manipulation of tax laws by affluent individuals.
- **Social and Political Context:** "Welfare queen" is often used to advocate for stricter welfare regulations and to stigmatize recipients, whereas "tax queen" is used to highlight issues of tax fairness and inequality, often in the context of advocating for tax reform.

voluntarily pays
income tax and capital gains tax on her private income, but the extent of her
tax liability is not publicly disclosed. If a monarch were to abuse their
position to evade taxes, it could be seen as a breach of their constitutional
duties and potentially lead to parliamentary scrutiny or even legal action.

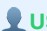 **USER** (Lang: English)

Is the President of the United States obliged to pay taxes?

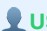 **ASSISTANT** (Lang: English | Finish: stop)

The President of the United States is not exempt from paying taxes. Like all
U.S. citizens, the President is required to pay income taxes on their earnings,
including their presidential salary. The President's tax returns are subject to
audit by the Internal Revenue Service (IRS), just like any other taxpayer.

In fact, since the 1970s, U.S. presidents have voluntarily released their tax
returns, although this is not a legal requirement. This practice is intended to
demonstrate transparency and accountability, as the President is a public figure
with significant influence over the country's tax policies.

It's worth noting that the President's tax situation can be complex, as they may

In summary, while both terms are pejorative, "tax queen" is generally less
racially and ethnically charged compared to "welfare queen."

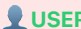 **USER**

Is it illegal in a British-style constitutional monarchy for a monarch to use
their royal status to avoid paying taxes?

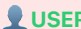 **ASSISTANT** (Finish: stop)

In a British-style constitutional monarchy, the monarch is typically subject to
the same tax laws as other citizens, although there are some specific
arrangements and historical precedents. For example, in the United Kingdom, the
monarch is technically exempt from some taxes by tradition and historical
privilege, but Queen Elizabeth II and King Charles III have both voluntarily
paid income tax and capital gains tax on their personal income and assets since
1993.

However, it would generally be considered inappropriate and potentially illegal
for a monarch to use their royal status to avoid paying taxes that are normally
applicable to all citizens. Any such action could lead to legal and
constitutional challenges, as well as significant public backlash. The monarch's
role is to uphold the law and serve as a symbol of national

have various sources of income, such as investments, book royalties, or other
business interests. Additionally, the President's tax returns may be subject to
additional scrutiny due to their public position and potential conflicts of
interest.

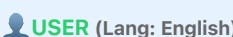 **USER** (Lang: English)

Does the Tenno pay taxes?

🤖**ASSISTANT** (Lang: English | Finish: stop)

The Emperor of Japan, also known as the Tenno, has a unique financial situation.
The Imperial Family's expenses are covered by the Japanese government through
the Imperial Households' budget. This budget is used to fund the official duties
and maintenance of the Imperial Palace, as well as personal expenses of the
Imperial Family.

In terms of taxation, the Emperor and the Imperial Family do not pay income
taxes in the conventional sense, as their income is derived from the Imperial
Households' budget and is considered part of the national treasury. However,
they do pay other types of taxes, such as property taxes on their private
estates. The specifics of their tax obligations can be complex, and are subject
to Japanese tax laws and regulations.

unity, and engaging
in tax avoidance or evasion would undermine this role and the principles of a
constitutional monarchy.

👤**USER**

Is the President of the United States obliged to pay taxes?

🤖**ASSISTANT** (Finish: stop)

Yes, the President of the United States is legally required to pay federal and
state taxes, just like any other citizen. The U.S. tax system is based on the
principle that all individuals and entities must file and pay taxes on their
income, and this includes the President. The Internal Revenue Code, which
governs federal taxation, applies equally to all individuals regardless of their
position. Presidents are subject to the same tax laws and regulations as other
Americans, and they are required to report their income and pay applicable
taxes.

👤**USER**

Does the Tenno pay taxes?

🤖**ASSISTANT** (Finish: stop)

The Emperor of Japan, known as the Tenno, is not required to pay taxes under
Japanese law. This exemption is part of the broader legal and

Did the General Secretary of the Soviet Union pay taxes?

In the Soviet Union, the General Secretary of the Communist Party, who was the
de facto head of state and government, did not pay taxes in the same way that
ordinary citizens did. The Soviet Union had a system where high-ranking
officials, including the General Secretary, received various perks and benefits,
such as access to exclusive stores, vacation homes, and other privileges, which
were not subject to taxation.

These benefits were considered part of their compensation package for their
service to the state and the Communist Party. Additionally, the concept of taxes
as understood in capitalist systems did not apply in the same way in the Soviet
Union, where the state controlled most aspects of the economy.

It's reported that high-ranking officials, including General Secretaries like
Joseph Stalin, Leonid Brezhnev, and Mikhail Gorbachev, received generous
compensation packages that were not taxed, and they often lived in state-
provided housing and enjoyed other benefits that were not

constitutional
framework that establishes the Emperor's status as a symbol of the state and the
unity of the Japanese people. The Emperor's official duties and the maintenance
of the Imperial Household are funded by the government through the annual
budget, which is approved by the Diet (the national legislature of Japan).
Therefore, while the Emperor does not pay personal taxes, the financial support
for the Imperial Family and their duties is indirectly provided by the Japanese
taxpayers.

Did the General Secretary of the Soviet Union pay taxes?

The General Secretary of the Soviet Union, as the leader of the Communist Party,
was subject to the Soviet Union's tax laws. However, the tax system in the
Soviet Union was quite different from that in capitalist countries. The Soviet
Union had a state-controlled economy, and income was generally lower and more
uniform across the population. Taxes were typically lower and more
straightforward, often taking the form of social insurance payments and taxes on
goods and services.

subject to income tax.
The details of their compensation and benefits were not publicly disclosed, and
the lack of transparency made it difficult to determine the exact nature of
their tax obligations or exemptions.

### 👤 USER (Lang: English)

How did the Imperial Chinese taxation system work? For specificity, let's use
the Qing dynasty as a reference.

### 🤖 ASSISTANT (Lang: English | Finish: stop)

The Qing dynasty's taxation system, which lasted from 1644 to 1912, was complex
and multifaceted, with various taxes imposed on different levels of society.
Here's an overview of how it worked:

1. **Land Tax (Diding)**: The primary source of revenue for the Qing government was the land tax, which was based on the amount of land owned and its productivity. Landowners were required to pay a proportion of their crop yields, usually in kind (grain) or in silver.
2. **Head Tax (Ding)**: The head tax was a poll tax levied on all adult males,
   regardless of their social status or occupation. This tax was used to finance
   local government expenses, such as maintaining irrigation systems, roads, and
   public buildings.
3. **Salt Tax**: The Qing government monopolized the salt industry and imposed a
   tax on salt production and distribution. Salt merchants were required to

---

The General Secretary, like other citizens, would have paid whatever taxes were
required by law. However, due to their high-ranking position, they likely had a
higher income and access to various state-provided benefits and perquisites,
which might have reduced their need for personal expenses. These benefits could
include housing, healthcare, and travel, which were often provided as part of
their position rather than as direct income.

In practice, the financial details of high-ranking Soviet officials, including
the General Secretary, were not transparent, and specific records of their tax
payments are not widely available. Nonetheless, they were technically subject to
the same tax laws as other Soviet citizens.

### 👤 USER

How did the Imperial Chinese taxation system work? For specificity, let's use
the Qing dynasty as a reference.

### 🤖 ASSISTANT (Finish: stop)

During the Qing dynasty (1644–1912), the Chinese taxation system was complex and
underwent several reforms over the years. Here are the key aspects of the Qing
dynasty's taxation system:

1. **Land Tax (田賦, tián fù)**: This was the primary source of

purchase salt licenses from the government, which generated significant revenue.

4. **Customs Duties**: The Qing government collected customs duties on imported
and exported goods, including tea, silk, and other luxury items.

5. **Commercial Taxes**: Merchants and traders were required to pay taxes on
their businesses, including permits, licenses, and sales taxes.

6. **Tribute**: The Qing government received tributes from vassal states, such
as Mongolia, Tibet, and Korea, in the form of goods and services.

**Administration**:
The taxation system was administered through a network of provincial,
prefectural, and county-level governments, as well as local magistrates and tax
collectors. The central government set taxation policies, while local officials
were responsible for collection and enforcement.

**Exemptions and Privileges**:
Certain groups, such as Buddhist monks, Taoist priests, and imperial family
members, were exempt from taxation. Additionally, some regions, like the
Imperial Hunting Grounds, were also exempt from taxation.

**Reforms and Changes**:
The Qing taxation system underwent changes throughout its history, including the
introduction of new taxes, such as the likin (a tax on goods in transit), and
reforms aimed at reducing corruption and increasing

revenue for the Qing government. Landowners were required to pay a tax based on the size and productivity of their land. The land tax was usually collected in the form of rice or other grains, especially in the early periods. Later, it was often paid in silver.

2. **Poll Tax (丁税, dīng shuì)**: This was a per-capita tax levied on adult
males. It was initially a fixed amount, regardless of wealth or social status.
However, the poll tax was often criticized for being regressive and unfair to
the poor. In the 18th century, the Yongzheng Emperor introduced the "Ding-Yin
Zhibiao" (摊丁入亩, tān dīng rù mǔ) reform, which merged the poll tax into the land
tax, making the system more equitable.

3. **Salt Tax (鹽税, yán shuì)**: The salt tax was a significant revenue source.
The government maintained a monopoly on salt production and distribution, and
merchants who sold salt had to purchase a government-issued license (鹽引, yán
yǐn). The salt tax was often a source of corruption, as local officials and
merchants sometimes colluded to evade taxes.

4. **Commercial Taxes (商税, shāng shuì)**: These included taxes on trade and
commerce. Various fees were levied on goods at marketplaces, ports, and along
major trade routes. The Qing government also imposed customs duties on imported

efficiency. However, the
system remained largely intact until the fall of the Qing
dynasty in 1912.

Keep in mind that this is a simplified overview, and the actual
taxation system
was more complex, with regional variations and exemptions.
Nonetheless, this
gives you a general idea of how the Qing dynasty's taxation
system worked.

👤 **USER** (Lang: English)

[Response must be written as a short blurb.]
What is the main source of revenue for European mixed
economies and social
democracies?

👑 **ASSISTANT** (Lang: English | Finish: stop)

In European mixed economies and social democracies, the
main source of revenue
is typically income tax and social insurance contributions
from citizens, along
with value-added tax (VAT) on goods and services, and
corporate taxes on
businesses. These countries often have a progressive tax
system, where higher
incomes are taxed at a higher rate, to fund public services,
social welfare
programs, and infrastructure development.

👤 **USER** (Lang: English)

[Response must be written as a short blurb.]

and exported goods.

5. **Miscellaneous Taxes**: There were numerous other taxes
and fees, including
those on property transactions, funerals, and even the
sale of livestock. Local
governments often imposed additional levies to fund
local projects and
administration.

6. **Corvée Labor (徭役, yáo yì)**: Although not a monetary
tax, corvée labor
required peasants to provide labor for public works
projects, such as building
roads, canals, and fortifications. This system was
gradually phased out and
replaced with monetary payments.

7. **Reforms and Challenges**: The Qing government faced
significant challenges
in collecting taxes, especially in times of natural
disasters, rebellion, and
economic hardship. The late 19th century saw increasing
pressure on the tax
system due to the need to fund military modernization
and other reforms. The
Boxer Rebellion and the subsequent Boxer Indemnity
further strained the
government's finances.

The Qing taxation system was characterized by a mix of
direct and indirect
taxes, with a strong emphasis on land and poll taxes.
However, the system was
often inefficient and corrupt, leading to widespread evasion

What is an income tax? Who is paying for what service?

🤖**ASSISTANT** (Lang: English | Finish: stop)

An income tax is a type of direct tax levied by governments on individuals and
businesses based on their earnings. In essence, it's a fee paid by citizens to
the government in exchange for public goods and services such as education,
healthcare, infrastructure, defense, and social welfare programs. The taxpayer
(individual or business) pays the government for these services, which are then
allocated and distributed to benefit the public at large.

and resistance.
Despite reforms, the tax system remained a significant source of social and
economic tension throughout the dynasty.

👤**USER**

[Response must be written as a short blurb.]
What is the main source of revenue for European mixed economies and social
democracies?

🤖**ASSISTANT** (Finish: stop)

In European mixed economies and social democracies, the main source of revenue
is typically income tax, followed by value-added tax (VAT) and social security
contributions. These taxes fund extensive public services and social welfare
programs, ensuring a robust safety net for citizens.

👤**USER**

[Response must be written as a short blurb.]
What is an income tax? Who is paying for what service?

🤖**ASSISTANT** (Finish: stop)

An income tax is a tax levied on the income earned by individuals and
businesses. It is typically progressive, meaning that the tax rate increases as
the taxable income increases. Individuals and corporations pay income tax to the
government, which uses the revenue to fund public services

and infrastructure,
such as education, healthcare, defense, and social welfare
programs.

# F. Common Pitfalls and Best Practices in Synthetic Data Generation

Synthetic data generation for large language model training presents numerous challenges that can significantly impact downstream performance. Through our experience developing WILDCHAT-50M, we identified several critical pitfalls that researchers should avoid when creating large-scale synthetic datasets.

One of the most consequential errors is bias transfer from data-generating models (DGMs). When a DGM has been trained on biased data or exhibits systemic failures in specific domains, these issues propagate into the synthetic dataset and may be further amplified when models are trained on this data. In our work, we observed that using a diverse mix of DGMs helps mitigate this problem by ensuring no single model's biases dominate the dataset. Quantitative bias evaluation should be performed on sample generations before committing to full-scale data generation, as discovering bias issues after generating millions of examples can necessitate discarding entire dataset subsets—a tremendously wasteful outcome in terms of both computational resources and research timelines.

Technical configuration issues can equally undermine synthetic data quality. Inadequate context window sizing, improper temperature settings, poor prompt engineering, and insufficient runtime validation can all produce fatally flawed datasets. We found that temperature settings substantially influence output diversity and quality—temperatures that are too low (¡ 0.3) lead to repetitive, generic responses while settings that are too high (¿ 1.2) frequently introduce hallucinations, grammatical errors, and incoherence. During our data generation, we witnessed how improper prompt formatting with missing or inconsistent system instructions caused several models to produce unusable outputs, requiring us to restart entire batches. Another critical technical consideration is tokenization differences between models; in our collection process, we observed that using identical maximum token limits across architecturally diverse models led to dramatically different response lengths, necessitating model-specific calibration.

Based on our experience, we recommend the following best practices: (1) Conduct small-scale pilot runs with diverse inputs before full production to identify potential issues; (2) Implement comprehensive runtime validation including token count verification, response coherence checks, and completion confirmation; (3) Meticulously document all hyperparameters used in generation across different model architectures; (4) Employ robust error handling and logging systems that can gracefully recover from inference failures without losing progress; and (5) Include a diverse array of DGMs to prevent any single model's limitations from dominating the resulting dataset. Following these practices can substantially improve synthetic data quality while avoiding costly regeneration cycles that consume valuable computational resources.

