# OpenReview forum: "WildChat-50M: A Deep Dive Into the Role of Synthetic Data in Post-Training"
_ICML.cc/2025/Conference — ICML 2025 poster_

### Official Review · Reviewer_XCnA · 2025-02-22

**Overall Recommendation:** 3

**Summary:**

This paper constructs a larger and more high-quality post-training chat dataset (called WildChat-50M) by getting responses to prompts from more than just one "data-generating model" (DGM). The authors get responses to prompts from the WildChat-1M dataset from 50 open-weight models. The dataset contains over 1M multi-turn conversations (2-3 turns on average). The authors curate a supervised fine-tuning dataset from WildChat-50M using a human-in-the-loop strategy and show that when Llama-3.1 8B is SFT-ed on it, it improves on 2 out of 9 benchmarks used for evalaution (length-controlled AlpacaEval and IF eval) over baselines (Tulu 3, Magpie Align, UltraChat). For the other 7/9 benchmarks, the model achieves similar performance as baselines, despite using 40% of the data used for Tulu 3. The authors show some additional empirical results, like that there is a strong effect of which DGM is used to construct the finetuning dataset on downstream performance.

## Update after rebuttal
Most of my points have been addressed (except the claim in the paper about teaching performance of models with better benchmark performance), but I remain my score as I still believe there are remaining questions about the broader usefulness/quality of rewild as a posttraining mixture.

**Claims And Evidence:**

Not all claims made in this submission are supported by clear and convincing evidence. The authors do a lot of experiments, which is good, but the paper would've been stronger if there were less experiments and each experiment was given more attention. Now, for many claims there are confounders that could explain the result. I will detail issues with each claim below.

**Main claim: Re-Wild is a strong SFT mixture**.
The claim that Re-Wild is a strong SFT mixture is based on fine-tuning one model on it and showing it improves over 3 baselines on 2 benchmarks, and stays the same for others or gets worse (for Math and MUSR). While this is still on average an improvement, it would be great to be more upfront about this in the text. The result is still strong in my opinion, given that you outperform on two benchmarks with 40% of the data. This experiment is essentially the only one that gives some insight into the quality of WildChat-50M over other SFT datasets, which leaves many questions open; how does Re-Wild do for other base models compared to existing SFT datasets, how does it interact with scale? Additionally, what is the performance improvement over simply using WildChat-1M?

**Blending DGMs doesn't benefit models**.
Although you can say that for the two blending runs you do it did not improve performance, you can't claim from that that it won't work in general. For this claim in the main paper: *"This finding indicates that SDQ depends primarily on prompt diversity, and it is most effective to optimize, rather than generalize, responses"*, you would require more experiments. All you can say now is that it does not always help to blend different DGMs, but it might for other blends.

**Models with strong benchmark performance are not better teachers for that benchmark**.
Again, there are confounders here, like maybe llama is a better teacher for another model from the same family, not because of benchmark performance (as you mention yourself further down the paper).

**LLMs that do not share pre- and post-training data still have very similar outputs**.
How do you know these randomly selected models do not share pre- and post-training data?

**Essential References Not Discussed:**

N.A.

**Experimental Designs Or Analyses:**

As mentioned in the claims section above, the experimental design has flaws, where sometimes claims are made that are not properly supported by the experiments due to confounders being present.

**Methods And Evaluation Criteria:**

Most methods and evaluation criteria make sense for the problem at hand, except one:

In order to investigate the effect of context window you truncate responses from a model with a larger context window. This does not make a lot of sense to me, as the truncated responses might not make sense anymore. Additionally, the average performance over benchmarks is not a good metric to look at in this experiment, and the authors do not report benchmark scores except the aggregate. Perhaps long-context is important for some not other and therefore truncating deteriorates performance on some not others? I would also like to see the full results of that experiment in the main paper as opposed to just the average over benchmarks.

**Other Comments Or Suggestions:**

- Your citation for Cohere command-r plus is a policy primer that has not much to do with the model itself and the authors you cite are not from cohere. Should probably just refer to the model announcement on their website.

- Nit: line 117-118l should remove the word "may" or change phrasing

- In Appendix D, it would be useful to refer to where those things can be found.

- Figure 3 and 4 are hard to interpret and do not add much, and it's completely unclear what the words meant in the original judgement by the models because the context is gone (as you mention yourself in line 366-375l)

- Table 6; would be useful if you can highlight the best score for each group of base LMs that are finetuned.

**Other Strengths And Weaknesses:**

**Strengths**
- The paper contributes an important dataset, which is a strong contribution given that there is a lot of opacity surrounding post-training datasets
- The paper does a lot of experiments aimed at understanding more about post-training on their dataset.

**Weaknesses**
All mentioned above in response to questions.

**Questions For Authors:**

- I'm confused about the pretrained versus post-trained model variants distinction; which models do you consider pretrained? You say you use 19 pretrained models (which I already wonder how they would respond to instructions out of the box unless theyve been trained on some instruction data in pretraining), and then in section 3.2 you call a bunch of post-trained models pretrained?

- Why release most of complete configuration files and not all? (line 124r)

- Why is DGM Q72 outperforming others on Math (table 2) but in Figure 1 the model trained on Re-Wild is not? Or is in table 2 the model trained only on WildChat-50M data and in figure 1 on the full Re-Wild split?

- Styling experiment: you refer to rows 1:4 but its unclear where.

**Relation To Broader Scientific Literature:**

The paper positions itself well within literature, but it's sometimes difficult to understand to what extent the contribution is an improvement over very related existing papers (like WildBench-1M), because they are not compared against. Some things can be described clearer w.r.t. prior research, e.g. it would be good to mention more clearly for the first experiment in section 3.1 that the base model trained is always llama 3.1 8B, also for the other baselines like tulu 3 etc (in the caption of Figure 1 for example).

**Theoretical Claims:**

N.A.

---

> ### Author Rebuttal · Authors · 2025-03-29
>
> We thank the reviewer for the thoughtful response. To the best of our ability, given the 5000 character limit, we address your comments and questions below.
>
> We agree that our claim about SDQ depending primarily on prompt diversity could be worded more carefully. In the camera-ready draft, we will re-scope the claim to be more in line with our findings, e.g., by inserting phrases such as “in our experimental setting”.
>
> When we wrote that LLMs “do not share pre- and post-training data”, we meant that they do not entirely share it (that at least some data differs between the models trained); we will add this qualifier to the camera-ready.
>
> To your point about being upfront about the performance of our mix in the text, the caption of figure 1 reads: “RE-WILD outperforms strong baselines, on average, across nine benchmarks. In particular, it exhibits strong performance on generalist chat and instruction following benchmarks.” We are glad your assessment that this is a strong result agrees with ours, hence, we refer to it as a strong mix.
>
> In Sec 2.2, we state: Sometimes we will not specify the SFT target model name; in that case, it will always be Llama-3.1-8B-Base := L8B. If you would like us to repeat this information in another section, we can.
>
> In Appendix D, we write “we have attached several relevant artifacts to this submission which would not have incorporated well into the body of the paper.” The artifacts in question are in the Supplementary Material ZIP attached to our submission.
> How does Re-Wild do for other base models compared to existing SFT datasets? This would be an interesting ablation – however, to conduct it, we would have had to retrain not only our own model but also every baseline, because the prior work does not. How we trained our baselines could in turn raise its own set of methodological questions. With limited resources, we feel consistency with prior work is sufficient.
>
> How does it interact with scale? We provide scaling experiments at 100k, 250k, and 500k in our paper (Fig. 2) and we have more in our repository, which we will share as part of the camera-ready.
>
> What is the performance improvement over simply using WildChat-1M? The improvement is significant; the results are available in our artifacts. We did not include this in the main paper out of respect for the WildChat-1M authors, who felt it would not be a fair comparison unless we improved their work by sampling every response from GPT-4, which was prohibitively expensive. However, we are prepared to reverse that choice and include it in the appendix if you feel it is vital.
>
> We cannot release all configs and logs because some of them were corrupted on our cluster. We will release all uncorrupted ones.
> In the camera-ready draft, we will be happy to remove the word “may” from line 117-118l, change the citation for Cohere command-r plus to the model announcement the Cohere website, and highlight the best score for each group of base LMs in Table 6.
>
> Regarding Figures 3 and 4, we will make available the detailed responses from the LLM judges in our repository associated with the camera-ready release – it is, therefore, a straightforward matter to recover the context around the words, should anyone wish to do so. We note, however, that many of these words are easily interpretable even out of context.
>
> To clarify the pretrained versus post-trained model variants distinction; all of our DGMs were post-trained. However, many of the post-trained variants started from the same pretrained checkpoint. Therefore, we have 19 unique pre-trained models (each of which is post-trained) and 35 post-trained model variants (with non-unique pre-trained models). We will add this clarification to our camera-ready.
>
> **Why is DGM Q72 outperforming others on Math (table 2) but in Figure 1 the model trained on Re-Wild is not?** This is a good question – the reason is that the baselines in Figure 1 are trained on different data than the models compared in Table 2. In particular, the Tulu 3 SFT mix, which outperforms ReWild SFT mix on MATH, contains significantly more math data in the mix.
>
> We will update the camera-ready draft to link to Table 3, rows 1:4; thanks for this correction.
>
> Thanks again for your time. In the event that you now feel more positively about our paper, we would appreciate it if you updated your score.

---

### Official Review · Reviewer_uUWy · 2025-02-28

**Overall Recommendation:** 2

**Summary:**

- The paper introduces WildChat-50M, the largest public chat dataset to date, featuring responses from 50+ different open-weight models (0.5B-104B parameters) participating in over 1M multi-turn conversations each.

- The authors created Re-Wild, a new supervised fine-tuning (SFT) data mix that outperforms Allen AI's Tulu-3 mixture while using only 40% as many samples.

- Key findings show that the choice of data generating model (DGM) significantly impacts downstream performance, sometimes more than model size or parameter count.

- The research demonstrates that SFT models inherit stylistic elements and response patterns from their DGMs, with LLM judge preferences transferring from DGM to fine-tuned model.

- Analysis reveals scaling laws for synthetic data, showing consistent performance improvements with larger dataset sizes.

- Technical investigations found high similarity between responses from different models, suggesting LLMs produce more predictable outputs than humans.

- The authors observed that models learn more effectively from DGMs in the same model family (approximate on-policy sampling).

- Blending responses from multiple DGMs showed no performance benefits; individual DGM quality was more important than diversity.

**Claims And Evidence:**

**Well-supported claims:**
- The RE-WILD data mix outperforming Tulu-3 with fewer samples is convincingly demonstrated in Figure 1 and through detailed benchmark comparisons across 9 different benchmarks.
- The claim that DGM choice significantly impacts downstream performance is well-supported by Table 2, showing substantial variance across different models.
- The scaling effects shown in Figure 2 provide clear evidence that performance improves with increased dataset size.

**Adequately supported claims:**
- LLM judge preferences transferring from DGMs to fine-tuned models is supported by Table 4 and word frequency analysis, though the selection of which judgments to analyze could introduce bias.

**Claims needing stronger evidence:**
- The assertion that Wildchat-50M is the "largest public chat dataset" lacks direct size comparisons with other datasets.
- The derivation of chat transcripts isn't fully explained.
- The claim about models learning better from same-family DGMs would benefit from more extensive cross-family experiments.

**Essential References Not Discussed:**

N/A.

**Experimental Designs Or Analyses:**

**Re-Wild Data Mix Evaluation:**
- Sound: Compared against established baselines (Tulu-3, Magpie, Ultrachat) on multiple metrics
- Issue: The composition of Re-Wild mix (Table 1) was "chosen heuristically" rather than through systematic optimization

**DGM Comparison Analysis (Table 2):**
- Sound: Controlled experiments across 6 models from 4 families (7B-104B parameters)
- Sound: Comprehensive evaluation across 9 benchmarks
- Issue: No statistical significance testing for performance differences between models, especially since some numbers seem quite close

**Scaling Experiments (Figure 2):**
- Sound: Consistent methodology across 100k, 250k, and 500k dataset sizes
- Sound: Tested with multiple DGMs to validate trends
- Issue: Did not extend to saturation points

**Style Inheritance Analysis (Table 3):**
- Sound: Quantitative measurement of stylistic elements using HTML tag frequency
- Issue: Limited to analyzing only 80 conversations from MTBench

**General Issues:**
- Limited target model diversity (primarily using Llama-3.1-8B Base)
- Fixed hyperparameters throughout experiments without extensive tuning
- Limited evaluation of generalization to different domains or specific prompt types
- General lack of indicators of statistical significance

**Methods And Evaluation Criteria:**

The methods and evaluation criteria are generally appropriate for studying synthetic data in post-training, with several strengths and some limitations:

**Strengths:**

- The benchmark selection balances LLM-judge metrics (MTBench, AlpacaEval2, MixEval) with ground-truth benchmarks (MMLU, BBH, GPQA), which mitigates biases inherent to either approach.

- The diverse set of 50+ models spanning different sizes and architectures supports generalizable conclusions about DGM quality.

- Their multi-faceted analysis (scaling laws, DGM comparison, style inheritance) effectively addresses different aspects of synthetic data quality.

**Limitations:**

- The evaluation focuses heavily on Llama-3.1-8B Base as the SFT target, which may not generalize to other model families or sizes.

- The response similarity analysis relies on automated metrics (ROUGE, METEOR) without additional evaluation to validate perceived similarity.

- The set of 9 benchmarks, while diverse, still primarily measures general capabilities rather than fully capturing the range of potential LLM applications.

- Limited analysis of how specific prompt types or domains benefit differently from various DGMs.

**Other Comments Or Suggestions:**

N/A

**Other Strengths And Weaknesses:**

**Strengths:**

- **Multi-faceted Analysis:** Comprehensive investigation combining style inheritance, scaling laws, and model comparison in a single study.

- **Effective Visualizations:** Figure 1's spider chart elegantly illustrates comparative model performance across multiple dimensions.

- **Practical Impact:** Re-Wild demonstrates that careful DGM selection can achieve better performance with fewer samples, democratizing high-quality model training.

**Weaknesses:**

- **Limited Theoretical Framework:** The paper lacks a cohesive theoretical model explaining why certain DGMs perform better than others.

- **Narrow Application Focus:** Focuses exclusively on SFT without exploring how findings might extend to RLHF or other post-training techniques.

- **Presentation Issues:** Some figures (like Figure 3-4) would benefit from quantitative scales rather than relative word sizes.

- **Terminology Inconsistency:** The paper introduces several abbreviations and naming conventions that can be difficult to track/understand without reading in depth.

**Questions For Authors:**

1. Did you conduct experiments varying only the DGM while keeping dataset size fixed versus varying dataset size while keeping the DGM fixed?

2. What theoretical explanation do you propose for why same-family models perform better as DGMs? Is it architectural similarity, tokenization consistency, or some other factor?

3. Your main experiments use Llama-3.1-8B as the target model. Did you perform any experiments with radically different architecture families (e.g., Mistral, Qwen) as targets to verify your conclusions generalize?

**Relation To Broader Scientific Literature:**

The paper's contributions connect to several established research threads in the LLM community:

- Extends the original WildChat dataset (Zhao et al., 2024) and parallels LMSys-Chat-1M (Zheng et al., 2024)

- Their DGM choice findings complement recent work on teacher-student model alignment

- Uses MT-Bench (Zheng et al., 2023) while examining biases in LLM judges (Feuer et al., 2024)

- Investigates scaling relationships specifically for synthetic data, complementing general scaling law research

**Theoretical Claims:**

I'm not aware of any theoretical claims made by this paper.

---

> ### Author Rebuttal · Authors · 2025-03-29
>
> We thank the reviewer for the thoughtful response. To the best of our ability, we briefly address each of your comments and questions below.
>
> To your point about confidence intervals; we agree that these would be valuable. Therefore, if our paper is accepted, we will add 95% confidence intervals using the normal approximation method to the main paper, where they are absent. We hope that this resolves your concern in this area.
>
> We agree that direct size comparisons with other datasets should be included. The largest such datasets we are aware of are WildChat-1M and LMSys-Chat-1M. Our dataset is more than 50x larger than either of those datasets. We will include this information in our camera-ready.
>
> We will improve our explanation of the derivation of chat transcripts by more thoroughly explaining the contributions of the original WildChat-1M, which included the acquisition of the human multi-turn conversations used in this work.
>
> We agree that our claim about models learning better from same-family DGMs would benefit from more extensive cross-family experiments; we will be happy to reword the claim to be more modest in scope in the camera-ready draft.
>
> The raw quantitative data used to generate Figures 3-4 will be made available in our repository for the camera ready release. To your point about the selection of which judgments to analyze potentially introducing bias, we agree, which is why we report results over all of MTBench without subselection.
>
> Our heuristic choice of SFT mix is consistent with prior work (Tulu 3); as of right now, automated methods for subselection trail human performance (Tulu 3), and it is a relatively low-cost annotation task for human experts.
>
> We agree that the response similarity analysis would benefit from a small-scale human study; if our paper is accepted, we will conduct such a study.
>
> We did conduct experiments varying only the DGM while keeping dataset size fixed; these can be found in our supplementary material ZIP attached to the submission.
>
> Sadly, at this time we can offer no theoretical basis for our findings; they are purely empirical. We consider this a useful direction for future work.
>
> We did perform experiments with Qwen as well as Llama; the results can be found in Appendix Table 6.
>
> To your point about RLHF; we agree with you that it would have been a valuable contribution, and we note as much in our limitations section; we consider this a very interesting and important direction for future work, and thank you for highlighting it. However, the paper, even in its current form, required considerable resources, and we hope you will take this into account.
>
> Thanks again for your time. In the event that you now feel more positively about our paper, we would appreciate it if you updated your score.

---

### Official Review · Reviewer_hY6u · 2025-03-19

**Overall Recommendation:** 4

**Summary:**

This paper constructed a large set of 50M synthetic conversations by using the initial human prompts from WildChat and various pretrained language models to generate responses and following turns. Based on the data, they further selected a subset and mixed with two other sources of data (MMLU Auxiliary Train and Tulu 3 Persona Hub Algebra) to construct form a dataset for instruction tuning base language models. This work also performed extensive analyses to study the effects of various choices in their dataset construction pipeline, such as whether blending different models helps.

### update after rebuttal
Authors mentioned that they'll consider adding a section addressing some of my concerns. I'm leaning positively about this paper and am keeping my score.

**Claims And Evidence:**

1. A critical question is whether the advantage of increased dataset scale (50x larger than the original WildChat) compensates for the potentially lower quality of synthetically-generated responses compared to human-written responses. The paper does not address whether this large-scale synthetic data genuinely surpasses smaller-scale, higher-quality human-written responses. I strongly suggest the authors conduct experiments to determine if synthetic data can effectively compensate for presumed quality deficits at larger scales. For example, the authors can add a comparison to Figure 2, where in addition to existing bars, add results for simply subsampling original WildChat to 100K, 250K, and 500K, and compare results. I'd hope to see that at smaller scales original data is better, but beyond a certain scale original data is no longer available and synthetic data finally catches up.
2. Section E Table 7 shows that blending responses from multiple language models yields no benefit (compared to using the strongest teacher). This result appears to undermine the motivation of exploring diverse synthetic responses, raising the question: is the primary benefit observed here due solely to the strong performance of a single DGM (Qwen 2.5 72B)?

**Essential References Not Discussed:**

N/A

**Experimental Designs Or Analyses:**

N/A

**Methods And Evaluation Criteria:**

1. Important details about how human turns within multi-turn conversations are generated remain unclear. Since pretrained language models typically generate only assistant responses, it is unclear how human dialogue turns were synthesized, how many turns were included per conversation, and whether such multi-turn interactions improve downstream performance. Clarifying these points would substantially strengthen the paper.
2. The ReWild dataset, although performing well, incorporates external datasets (MMLU Auxiliary Train, Tulu 3 Persona Hub Algebra), somewhat diluting the primary contribution of the WildChat-50M dataset itself.

**Other Comments Or Suggestions:**

On page 4 there's too much white space.

**Other Strengths And Weaknesses:**

N/A

**Questions For Authors:**

Please see above.

**Relation To Broader Scientific Literature:**

WildChat-50M is a substantial resource, significantly expanding the availability of synthetic conversational data for training or fine-tuning language models, greatly benefiting the research community.

**Theoretical Claims:**

N/A

---

> ### Author Rebuttal · Authors · 2025-03-29
>
> We thank the reviewer for the thoughtful response. To the best of our ability, we briefly address each of your comments and questions below.
>
> Regarding experiments to determine if synthetic data can surpass the performance of smaller-scale, higher-quality human-written chat responses: unfortunately, we know of no such publicly available chat dataset. The original WildChat-1M dataset contained GPT 3.5 and GPT-4 synthetic responses (we did train models on that data; the results of those experiments are available in our artifacts). The closest comparisons of which we are aware are the large-scale post-training runs exclusively on FLAN data described in (https://arxiv.org/abs/2409.15268). FLAN has human prompts and human responses, but is not a chat dataset; that work found that FLAN-trained models significantly underperformed WildChat-trained ones, controlling for dataset size. We agree that more research needs to be done in this area.
>
> As to whether the primary benefit observed in our mix is due solely to the strong performance of a single DGM (Qwen 2.5 72B), our key contributions are (1) discovering which DGMs tend to produce higher quality responses and (2) providing experimental evidence on why they perform better.
>
> We agree that we could have more thoroughly explained the contributions of the original WildChat-1M paper, which included the acquisition of the human multi-turn conversations used in this work. We will add more information on this to our camera-ready draft.
>
> For the camera-ready version, we will also correct the issue of too much white space on page 4.

---

> > ### Comment · Reviewer_hY6u · 2025-04-07
> >
> > Sorry for the confusion in my wording, but I meant compared to chatgpt generated data. For example, why would wildchat-50M be useful when there's WildChat-1M? Can quantity (50M as opposed to 1M) overcome quality (presumably at least the GPT-4 portion of WildChat has higher response quality than WildChat-1M)?

---

> > > ### Author Response · Authors · 2025-04-08
> > >
> > > Thanks for the response! This is a very intriguing question. While we did not have enough GPT-4 data in WildChat-1M to do extremely large-scale comparisons, we did conduct small-scale comparisons. In those, it did not appear to be the case that GPT-4 as a DGM was more helpful than the best open-weights models we tested. This surprising result makes more sense in the light of some of our other findings about how LLMs actually learn from DGMs; style rather than factuality. If the paper is accepted, we are happy to add a section discussing this to the appendix, if you wish.

---

### Official Review · Reviewer_86kE · 2025-03-21

**Overall Recommendation:** 3

**Summary:**

Authors are proposing a new synthetic dataset: 'WILDCHAT-50M'
Compare to other open datasets, 'WILDCHAT-50M' is much larger and includes synthetic data generated from many open source models other than GPT.

-WILDCHAT-50M is the largest public chat dataset to date
-It includes responses from over 50 different open-weight models
-A comparative analysis was conducted using this dataset
-RE-WILD, a public SFT mix, was created and outperformed a recent mixture from Allen AI

**Claims And Evidence:**

Claim - The choice of DATA Generative Model (DGM) significantly impacts downstream model performance on generalist chat benchmarks. Selecting a good DGM can compensate for small dataset size and outperform more complex methods and carefully curated SFT mixes.
Evidence:  compare the
performance of six unique pretrained models from four distinct model families, including Qwen-2.5-72B-Instruct from
Alibaba, Llama-3.3-70B-Instruct from Meta, Command-RPlus from Cohere, and Jamba-1.5-Mini from AI21.
Benchmarked on:  MTBench AlpacaEval BBH GPQA MATH MUSR IFEval MMLU Pro MixEval
Showed that the results have large variance and is unpredictable.
- Claim: There is no benefit in generating data generation models. Certain DGMs produce higher Synthetic Data Quality (SDQ) due to factors such as: Comprehensiveness, Clarity, Tone, Prompt responsiveness. These factors are highly heritable during the SFT process, even on generalist data.
- Skills like world knowledge or mathematics are only heritable when data is curated for that specific purpose.
- Large Language Models (LLMs) exhibit a high degree of similarity in prompt responses, suggesting a subtle distinction between high and low SDQ.

**Essential References Not Discussed:**

N/A

**Experimental Designs Or Analyses:**

The experiment setups are straight forward.
Training framework: Axolotl
Eval framework: Evalchemy
With all standard settings

**Methods And Evaluation Criteria:**

For benchmarking, author used a few different benchmarks.

MTBench AlpacaEval BBH GPQA MATH MUSR IFEval MMLU Pro MixEval

**Other Comments Or Suggestions:**

Check wording:

 When we fine-tune Llama-3.1 8B Base on RE-WILD, 'wehow' that our models outperform the SFT mix proposed in Tulu-3

Should be 'we show'? I suggest use grammarly.

**Other Strengths And Weaknesses:**

Strength:
- The authors presented a large synthetic dataset that is benefit for the open source community.

Weakness:
- Most of the claims in the paper are well established by past publications. There is little novelty in the claims.
- Limited post-training approaches: Only SFT (Supervised Fine-Tuning) was used, and results may differ with other post-training methods.
- Benchmark suite limitations: The benchmark suite is standardized, balanced, and large, but does not cover all use cases, particularly:
Highly specialized tasks (e.g., coding, legal reasoning)

**Questions For Authors:**

N/A

**Relation To Broader Scientific Literature:**

Post-training techniques for language models: The paper builds upon prior work on post-training techniques, such as Supervised Fine-Tuning (SFT) , which has been shown to improve the performance of language models. The authors' contribution lies in exploring the effect of different Data Generating Models (DGMs) on SFT.
Importance of dataset diversity: The paper's focus on creating a large and diverse dataset (WILDCHAT-50M) is in line with previous research highlighting the importance of dataset diversity for training robust language models. The authors' contribution is in providing a standardized benchmark suite that can be used to evaluate the performance of different DGMs.
Comparative analysis of language models: The paper's comparative analysis of different DGMs is similar to previous studies that have compared the performance of various language models on specific tasks . However, the authors' contribution is in providing a comprehensive evaluation of multiple DGMs on a large and diverse dataset.
Open science and reproducibility: The paper's commitment to open science and reproducibility is in line with recent efforts to promote transparency and accountability in AI research. The authors' decision to release their data, artifacts, and code publicly aligns with these values.

**Theoretical Claims:**

There are no theoretical claims in this paper.

---

> ### Author Rebuttal · Authors · 2025-03-29
>
> We thank the reviewer for the thoughtful response. To the best of our ability, we briefly address each of your comments and questions below.
>
> You are correct that "wehow" is a typo and should read “we show”. We apologize for the inconvenience, and will remedy this in the camera-ready draft if we are accepted.
>
> You are also correct that our benchmark suite contains no examples of highly specialized coding or legal reasoning tasks. This exclusion was motivated by our findings about the limited usefulness of generalist chat data for such tasks (see our conclusions at the end of 3.3).
>
> To your point about our inability to include other post-training approaches in this work, we agree with you that it would have been a valuable contribution, and we note as much in our limitations section; we consider this a very interesting and important direction for future work, and thank you for highlighting it. However, the paper, even in its current form, required considerable resources, and we hope you will take this into account.
>
> Thanks again for your time. In the event that you now feel more positively about our paper, we would appreciate it if you updated your score.

---

### Official Review · Reviewer_a9Br · 2025-03-23

**Overall Recommendation:** 3

**Summary:**

The paper proposes a new synthetic chat dataset, WildChat-50M, which consists of generated responses from 50+ open weight models. The authors then created a new SFT datamix, Re-Wild, by combining WildChat-50M with two other datasets (MMLU Auxiliary Train, Tulu 3 Persona Hub Algebra). Main contribution of the paper:
- Introduction of new datasets WildChat-50M and Re-Wild and their source codes
- Ablation showing its SFT performance beats other open datamix on 9 benchmarks
- Analysis on the effect of data generating models (DGM) efficiency and their impact on the synthetic data quality (SDQ), and hypothesis why certain DGMs outperforms the rest on certain benchmarks
- Misc empirical insights on what tricks matter and what don't, e.g. choice of DGM, diversity of DGM, context window, ...

**Claims And Evidence:**

The major claim of the paper:
- WildChat-50M is a useful dataset, evidenced by its derived SFT datamix, Re-Wild outperforming other open SFT datamix baselines in finetuning llama3 8b.
- Various factors contributed to its source of effectiveness, including data volume, DGM category, etc.
- The choice of DGM is critical and highly diversified across different benchmarks.

Evidence:

Figure 1-2 showing that
- Overall Re-Wild outperforms strongly over other baselines.
- SFT performance improves as the data volume increases.

Table 2 showing that the choice of DGM differs drastically across benchmarks

**Essential References Not Discussed:**

n/a

**Experimental Designs Or Analyses:**

see above

**Methods And Evaluation Criteria:**

This is an empirical paper. It is well written and straightforward to follow through. The figures and tables are well presented and the introduction, related work provide thorough context on the related literature. While the paper is not technically novel, I'm leaning toward acceptance because
- It introduces a new SFT dataset that's open source and useful
- The design of this datamix is backed by extensive study and comprehensive analysis is included to inform the reader of key design choices
- Efficiency is included and key parameters are transparently disclosed

Thoughts on evaluation
- While the reviewer acknowledges the performance gain after SFT, how well it carries over to RLHF is unknown -- would a stronger SFT base from finetuning on this synthetic data result in a stronger RLHF candidate? In an industry setup, usually SFT is followed by RLHF and the model quality after RLHF is what truly matters.

Thoughts on ablations
- The SFT datamix shows strong performance -- and there might have been considerable iteration cycles before it finally worked. The reviewer imagines the synthetic data generation to not be something straightforward and there are many decisions crucial to the success. Would be helpful to callout any common fallacies and summarize the best practices in synthetic data generation, e.g. is there some config that is easily overlooked but would ruin the entire dataset if not carefully engineered. Such empirical insights would add great benefits to the industrial community.

Minor issues
- Tab 1 / row 1 -- should it be WildChat-50M instead of WildChat-Q72?

**Other Comments Or Suggestions:**

n/a

**Other Strengths And Weaknesses:**

n/a

**Questions For Authors:**

see above

**Relation To Broader Scientific Literature:**

The following are related:
- LLM post training
- supervised finetuning
- synthetic data generation

**Theoretical Claims:**

no theoretical claim was proposed

---

> ### Author Rebuttal · Authors · 2025-03-29
>
> We thank the reviewer for the thoughtful response. To the best of our ability, we briefly address each of your comments and questions below.
>
> To your point about our inability to include RLHF in this work (and therefore to answer your question about whether stronger SFT leads to stronger post-trained models), we agree with you that it would have been a valuable contribution, and we note as much in our limitations section; we consider this a very interesting and important direction for future work, and thank you for highlighting it. However, the paper, even in its current form, required considerable resources, and we hope you will take this into account.
>
> We agree that a section documenting common pitfalls and best practices in synthetic data generation, including configurations that might ruin the entire dataset, would be valuable; we do not include it here because of space constraints, however, if the paper is accepted, we will add such a section to the appendix in the camera-ready version.
>
> We are happy to confirm that Table 1, Row 1 is in fact not a typo – the name refers to samples with WildChat-1M prompts and Qwen 2.5 72B Instruct responses, following the convention introduced in Sec. 2.2. However, as we only explicitly define naming conventions for models and here implicitly use it to name a dataset, there could be confusion -- we will clarify that the names can apply to datasets generated using DGMs in the camera-ready draft.
>
> Thanks again for your time. In the event that you now feel more positively about our paper, we would appreciate it if you updated your score.

---

### Decision · Program_Chairs · 2025-05-01

**Decision:**

Accept (poster)

**Comment:**

This paper introduces WildChat-50M, a synthetic dataset generated by 50+ open-weight models, and Re-Wild, a supervised fine-tuning (SFT) datamix that outperforms existing baselines with fewer samples. Key contributions include empirical insights into
1) impact of data-generating model (DGM) on performances,
2) data scaling
3) models inherit stylistic patterns from DGMs

While reviewers praised the dataset’s scale and practical utility, concerns centered on
1) Limited comparison to human-written data (e.g., those from WildChat) to validate synthetic data quality.
2) more evidence needed around the claim about mixing DGM, especially when some are shown superior to others as claimed in the paper.
3) lack of details on multi-turn conversation synthesis (human vs. model-generated turns).
4) generalizability of the claims to other model sizes/families. All experiments validating this new synthetic dataset is based on supervised finetuning a llama3 8b model.

WildChat-50M and Re-Wild provide valuable resources for the LLM community. The paper also offers valuable insights into SFT data and DGM choices.

However there are also some concerns (including those highlighted above) not fully addressed and remain open. Given that this work offers a foundation for open research on data drive post training, I'm leaning towards acceptance when all the concerns listed above are fully addressed.